# OccSora: 4D Occupancy Generation Models as World Simulators for Autonomous Driving

## Abstract

Understanding the evolution of 3D scenes is important for effective autonomous driving. While conventional methods model the scene development with the motion of individual instances, world models emerge as a generative framework to describe the general scene dynamics. However, most existing methods adopt an autoregressive framework to perform next-token prediction, which suffer from inefficiency to model long-term temporal evolutions. To address this, we propose a diffusion-based 4D occupancy generation model, OccSora, to simulate the development of the 3D world for autonomous driving. We employ a 4D scene tokenizer to obtain compact discrete spatial-temporal representations for 4D occupancy input and achieve high-quality reconstruction for long-sequence occupancy videos. We then learn a diffusion transformer on the spatial-temporal representations and generate 4D occupancy conditioned on a trajectory prompt. We conduct extensive experiments on the widely used nuScenes dataset with Occ3D occupancy annotations. OccSora can generate 16s videos with authentic 3D layout and temporal consistency, demonstrating its ability to understand the spatial and temporal distributions of driving scenes. With trajectory-aware 4D generation, OccSora has the potential to serve as a world simulator for the decision-making of autonomous driving.

## 1 Introduction

As a critical application of artificial intelligence technology, autonomous driving has garnered widespread attention and research in recent years (Hu et al., 2023b; Fu et al., 2024; Yang et al., 2023a). Establishing the relationship between perception (Liu et al., 2023c; Chang et al., 2023; Chen et al., 2023; Mao et al., 2023), prediction, and planning (Mozaffari et al., 2020; Huang et al., 2023b; Jia et al., 2023; Wang et al., 2024b) in autonomous driving is crucial for a comprehensive understanding of the field.

Conventional autonomous driving models (Hu et al., 2023b) rely on the motion of the ego vehicle instances to model the development of scenes, unable to develop a profound understanding of scene perception and vehicle motion control comparable to human understanding. The emergence and establishment of world models (Ha and Schmidhuber, 2018) offer new possibilities for a deeper understanding of the comprehensive relationship between autonomous driving scenes and vehicle motion. Based on strong image pretrained models, image-based world models (Hu et al., 2023a; Wang et al., 2023b) are able to generate high-quality driving-scene images with conditions of 3D bounding boxes. OccWorld (Zheng et al., 2023) further learns a world model in the 3D occupancy space, which can be better leveraged for 3D reasoning for autonomous driving. However, most existing methods adopt an autoregressive framework to model the dynamics (e.g., image tokens, bounding boxes, occupancy) of a 3D scene, and thus cannot efficiently produce long-term sequences.

To address this, we propose a 4D world model OccSora to directly generate spatial-temporal representations with diffusion models as shown in Figure 1, motivated by OpenAI's 2D video generation model Sora (Brooks et al., 2024). To accurately understand and represent 4D scenes, we design 4D scene discretization to capture the dynamic characteristics of scenes and propose a diffusion-based world model to achieve controllable scene generation in accordance with physical laws. Specifically, in the 4D occupancy scene tokenizer, we focus on extracting and compressing real 4D scenes to establish an understanding of the world model environment. In the diffusion-based world model, we employ multidimensional diffusion techniques to propagate accurate spatiotemporal 4D information

Figure 1: **Comparisons with existing methods.** It can comprehend the intricate relationship between scenes and trajectories and generate long-term, physically consistent 4D occupancy.

and realize trajectory-controllable scene generation by incorporating real ego car trajectories as supervision, thereby achieving a deeper understanding between autonomous driving scenes and vehicle motion control. Through training and testing, OccSora can generate autonomous driving 4D occupancy scenes that adhere to physical logic and achieve controllable scene generation based on different trajectories. The proposed autonomous driving 4D world model opens up new possibilities for understanding dynamic scene changes in autonomous driving and the physical world.

## 2 RELATED WORK

**3D Occupancy Prediction.** 3D occupancy focuses on partitioning space into voxels and assigning specific semantic types to each voxel. It is considered a crucial means of representing real-world scenes, following 3D object detection (Mao et al., 2023; Ma et al., 2023; Yu et al., 2024) and Bird's Eye View (BEV) perception (Yang et al., 2023b; Zhao et al., 2024; Wang et al., 2023a; Zhang et al., 2022), for autonomous driving perception tasks. Early research on this task primarily focused on semantically classifying discrete points from LiDAR (Zhou et al., 2021; Singh et al., 2020; Liu et al., 2023a). In fact, due to the camera containing semantic information far exceeding that of LiDAR and their low cost. Thus, utilizing images for depth estimation or employing end-to-end methods for 3D scene perception research is currently the mainstream approach (Huang et al., 2023a; Li et al., 2023; Wei et al., 2023). Considering the advantageous of multi-sensor systems, some studies research multi-modal fusion for 3D occupancy prediction (Wang et al., 2023c; Zhang and Ding, 2024).

In addition to utilizing typical sensor devices for 3D occupancy prediction, some studies focus on other tasks involving occupancy. For instance, OccWorld (Zheng et al., 2023) proposes a spatiotemporal generative transformer to predict subsequent scene tokens and the vehicle token, thereby predicting future occupancy and vehicle trajectory. On the other hand, GenOcc (Wang et al., 2024a) utilizes generative models to accomplish occupancy prediction. DriveWorld (Min et al., 2024) introduces a world-model-based framework for learning in autonomous driving from 2D images and videos, addressing tasks such as 3D object detection, online map creation, and occupancy prediction. Although progress has been made in 3D occupancy prediction and continuous 4D prediction, the scope of these studies remains limited. They usually use autoregressive models in conjunction with scene information from preceding frames to carry out subsequent occupancy tasks, thereby necessitating prior scene or 3D bounding box inputs. Besides, the performance drop over time is significant in both 4D occupancy forecasting and motion planning, demonstrating the struggle of autoregressive models for generation in the autonomous driving field. This is because the complexity of capturing the spatial and temporal information of vehicle movements and scenes is greater than that of language text. Consequently, they lack a genuine understanding of the fundamental relationships between scene and motion, and they do not constitute world models conditioned on actions.

**Generative Model.** Generative models have garnered widespread attention recently due to their powerful capabilities. By learning the probability distribution of data, generative models can train models capable of generating new samples. From the emergence of Generative Adversarial Networks (GAN) (Goodfellow et al., 2020) to the recent advent of diffusion models like Variational Autoencoders (VAE) (Van Den Oord et al., 2017), the tasks of generative models have gradually expanded from initial image generation tasks to in-depth studies on videos (Yan et al., 2021). Tasks such as image generation based on the DIT model (Peebles and Xie, 2023) delve into and utilize their generative capabilities. The Sora video generation model (Brooks et al., 2024) further demonstrates the ability to produce high-quality videos with realistic transitions between frames in continuous scenes.

Similarly, in the field of autonomous driving, controllable image generation can provide various driving scenarios to serve perception, planning, control, and decision-making tasks. For instance, MagicDriver (Gao et al., 2023) generates videos depicting various weather scenarios by learning

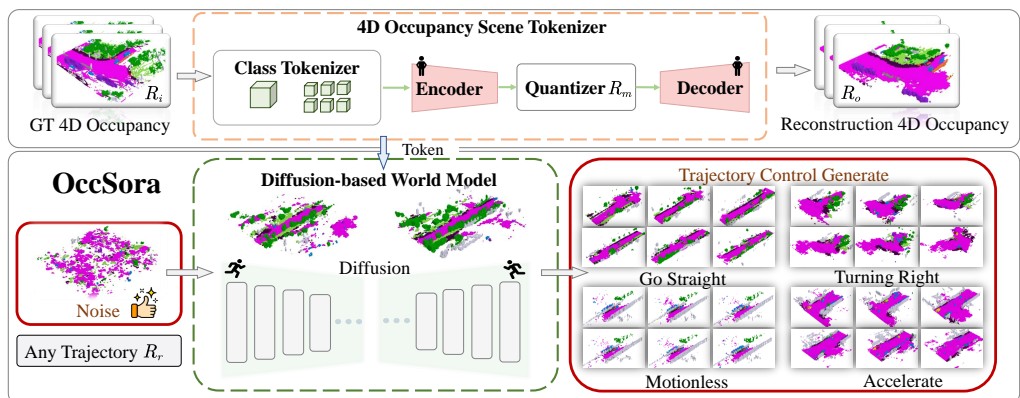

Figure 2: **The pipeline of OccSora.** The 4D occupancy scene tokenizer achieves compression and restoration of real information. The compressed information and vehicle trajectories are simultaneously used as inputs for the diffusion-based world model, generating trajectory-controllable tokens that are decoded into 4D occupancy.

from videos of autonomous driving vehicles and incorporating labels such as object detection boxes and maps. DriveDreamer (Wang et al., 2023b) proposes a world model that is entirely derived from real-world driving scenes, enabling a deep understanding of structured traffic constraints and thereby achieving precise and controllable video generation. However, for autonomous driving scenarios, obtaining the 3D occupancy of scenes is more important compared to 2D information (Zhang et al., 2023b; Mescheder et al., 2019; Sima et al., 2023). Some studies (Lee et al., 2024; Liu et al., 2023b) propose a three-dimensional diffusion model suitable for generating outdoor real scenes, which, by utilizing diffusion methods, accomplishes scalable seamless scene generation tasks. While some previous studies have generated 2D static images and extended them to the temporal dimension through autoregression, and others have achieved static generation of 3D occupancy scenes, both the 2D images generated based on 3D object bounding boxes and the static large-scale scenes are difficult to directly apply to autonomous driving tasks (Wang et al., 2024b; Zheng et al., 2024). In contrast, our proposed OccSora establishes a dynamic 4D occupancy world model that adapts to scene changes with vehicle trajectories, without the need for any prior object detection boxes or scene information, representing the first generative 4D occupancy world model for autonomous driving.

## 3 OCCSORA

### 3.1 WORLD MODEL FOR AUTONOMOUS DRIVING

Considering that 4D occupancy can comprehensively capture the structural, semantic, and temporal information of a 3D scene and effectively facilitate weak supervision or self-supervised learning, it can be applied to visual, LiDAR, or multimodal tasks. Based on these principles, we represent the world model $\chi$ as 4D occupancy $R$. Figure 2 illustrates the overall framework of OccSora. We constructed a 4D occupancy scene tokenizer (Van Den Oord et al., 2017) to compress real 4D occupancy $R_i \in \mathbb{R}^{B \times D \times H \times W \times T}$ in both the temporal $T$ and spatial $D \times H \times W$ dimensions, capturing the relationships and evolution patterns in 4D autonomous driving scenes. This results in compressed high-level tokens $R_m \in \mathbb{R}^{B \times c \times h \times w \times t}$ and reconstructed 4D occupancy data $R_o \in \mathbb{R}^{B \times D \times H \times W \times T}$. We designed a diffusion-based world model that uses trajectory information $R_r \in \mathbb{R}^{B \times T \times 2}$ as control units, training them along with the compressed tokens $R_m$ to generate high-dimensional scene representation tokens $T_o \in \mathbb{R}^{B \times c \times h \times w \times t}$. These are then decoded by the 4D occupancy scene tokenizer into physically consistent and dynamically controllable $R_o$.

### 3.2 4D OCCUPANCY SCENE TOKENIZER

The goal of 4D occupancy prediction is to determine the semantic type at specific locations over time. We discretize and encode the real 4D occupancy scene $R_i$ into an intermediate latent space $R_m$ to obtain a true representation of the 4D occupancy scene, as shown in Figure 3. The formula is as follows: $R_m = \zeta_{token} \{ \tau_{en} (R_i) \}$. Here, $\zeta_{token}$ represents the encoded codebook, and $\tau_{en}$ denotes the designed 3D encoder network and category embedding. This 3D occupancy representation divides the 3D space around the vehicle into voxels $r^T = N \in \mathbb{R}^{H \times W \times D}$, where each voxel position is assigned a

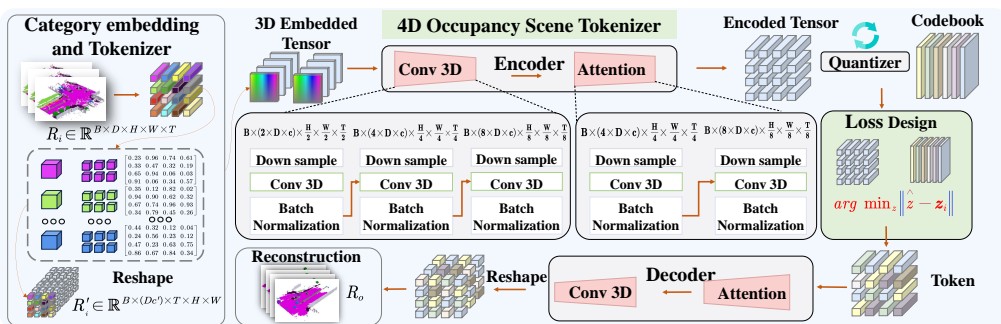

Figure 3: **The structure of the 4D occupancy scene tokenizer.** The proposed method encodes and compresses 4D scenes to extract high-dimensional features, which are then decoded to retrieve the spatiotemporal physical characteristics of the scenes.

type label $T_l$, indicating whether it is occupied and the semantics of the object occupying it. Unlike traditional methods, we incorporate and compress temporal information within the same scene, reshaping the tensor to $R_i$. This approach allows for unified learning of both spatial and temporal evolution patterns and the physical relationships of real scenes, compared to previous autoregressive methods. After passing through the $\tau_{en}$ 3D encoder network with category embedding and the $\zeta_{token}$ encoded codebook, the tensor is transformed into $R_m$ representing the potential spaces. This reshaping ensures a comprehensive representation of the temporal dynamics of 4D occupancy.

**Category embedding and Tokenizer.** To accurately capture the spatial information of the original parameters, we first perform an embedding operation on the input $R_i$. For each category in $R_i$, we assign a learnable category embedding $b \in \mathbb{R}^{c'}$, which serves to label the categories of continuous 3D occupancy scenes with dimensions $\mathbb{R}^{B \times D \times T \times H \times W \times c}$. The positional information is embedded as tokens representing the categories, resulting in dimensions $\mathbb{R}^{B \times D \times c \times T \times H \times W}$. These embeddings are then concatenated along the feature dimension. To enable subsequent 3D encoding with dimensional compression, $R_i$ is further reshaped into $R_i' \in \mathbb{R}^{B \times (Dc') \times T \times H \times W}$.

**3D Video Encoder.** To effectively learn discrete latent tokens, we further performed downsampling on the embedded positional information of the 4D occupancy $R_i'$ to extract high-dimensional features. The designed encoder architecture comprises a series of 3D downsampling convolutional layers, which perform 3D downsampling in both the time dimension (T) and spatial dimensions (H × W), increasing the fusion dimension to $D \times c'$. We initially downscaled the input $R_i'$ three times to obtain $R_i'' \in \mathbb{R}^{B \times (8 \times Dc') \times \frac{T}{8} \times \frac{H}{8} \times \frac{W}{8}}$, and introduced dropout layers after the feedforward and attention block layers for regularization. Considering the relationships between consecutive frames, we introduced cross-channel attention after downsampling, segmenting $R_i''$ along the $8 \times Dc'$ dimension and then performing cross-channel attention between the segmented parts. This operation enhanced the model's ability to capture relationships between features along different axes, and subsequently reshaped them back to the original shape to obtain the output tensor $R_m$.

**Coodbook and Loss Design.** To achieve a more condensed representation, we simultaneously learn a codebook $\zeta_{token} \in \mathbb{R}^{N_c \times D}$ containing $N_c$ codes. Each code $b \in \mathbb{R}^{c'}$ in the codebook encodes a high-level concept of the scene, such as whether the corresponding position is occupied by a car. $\zeta_{token}$ represents the encoded codebook. We quantize each spatial feature $\widehat{R_m^{(ij)}}$ in $\widehat{R_m}$ by mapping it to the nearest code $N(\widehat{R_m^{(ij)}}, B)$:

$$R_m^{(ij)} = N(\widehat{R_m^{(ij)}}, \zeta_{token}) = \min_{b \in \zeta_{token}} ||\widehat{R_m^{(ij)}} - b||_2, \quad (1)$$

where $|| \cdot ||_2$ represents the L2 norm. Subsequently, we integrate the quantized features $\widehat{R_m^{(ij)}}$ to obtain the final scene representation $R_m$.

**3D Video Decoder.** To reconstruct $R_o$ from the learned scene representation $R_m$, we design a decoder consisting of 3D deconvolution layers. In contrast to the encoder, the decoder architecture includes cross-channel attention, residual blocks, and a series of 3D convolutions, enabling upsampling in both temporal and spatial dimensions. This gradual upsampling process transforms $R_m$ to its

Figure 4: **The structure of the diffusion-based world model.** The model involves utilizing the optimal codebook obtained from training the 4D occupancy scene tokenizer to convert 4D occupancy into a sequence of tokens. These tokens, along with the ego vehicle trajectory and random noise, are then combined as input for denoising training to acquire the generated token.

original occupancy resolution $R_o$. The decoder then splits the result along the channel dimension to reconstruct the temporal dimension, yielding occupancy values for each voxel. During training, we accomplished the training of the encoder, decoder parameters, and the encoding codebook. The designed network enables us to simultaneously encode the input 4D occupancy information and compress it into multiple tokens, thereby learning the physical correlations of world models under spatiotemporal fusion. Additionally, we restore the information during the decoding process.

### 3.3 DIFFUSION-BASED WORLD MODEL

Inspired by the diffusion method (Peebles and Xie, 2022), we use scene tokens $R_m$ containing spatiotemporal information features as inputs for the generative model. Additionally, we conduct denoising training and trajectory-controllable generation tasks under the control of vehicle trajectories $R_r$, as shown in Figure 4.

**Token embedding.** To efficiently and accurately utilize the transformer (Vaswani et al., 2017), we flatten the input data tokens $R_m$ into $R_{\text{re}}$. Simultaneously, considering the significance of positional information for spatiotemporal compression, we perform positional embedding on the input. We design the following function, which utilizes sin and cos functions to encode positional indices:

$$R_{re}^{(\text{emb})} = \text{emb}_i^d + R_{re}, R_{re} \in \mathbb{R}^{B \times c \times (hwt)}. \tag{2}$$

It operates on two main parameters: $C$, representing the embedding output dimensionality of each position, and $i = hwt$, representing the number of tokens enumerating the positions to be encoded. The resulting output follows a matrix structure of dimensions $C \times i$, and *emb* constructs the positional embedding representation using sin and cos functions. These embeddings encapsulate the positional attributes of the tokens, enhancing the model's understanding of positions within the input. We add the positional encoding *emb* to the input $R_{re}$, yielding $R_{re}^{(\text{emb})}$, which represents the tokens after positional encoding.

**Trajectory conditioning embedding.** The transformation relationship between scenes and trajectories is a crucial aspect of autonomous driving. Generating diverse 4D occupancy scenes that align with control trajectories is essential. Therefore, we use the ego vehicle trajectory $T_r$ as input to generate controllable 4D occupancy. Firstly, the ego vehicle trajectory $T_r \in \mathbb{R}^{B \times t \times 2}$ is used as one of the control inputs, where $t$ denotes the continuous time dimension, and the third dimension represents the vehicle positions along the $x$ and $y$ axes of the absolute coordinate system. To achieve trajectory embedding and encoding, we reshape the vehicle trajectory to $T_r \in \mathbb{R}^{B \times (t \times 2)}$ and learn and encode it as follows:

$$g = \nu(t_d) + \delta(T_r), \delta(T_r) \in \mathbb{R}^{B \times c \times (hwt)}, \nu \in \mathbb{R}^{B \times c \times (hwt)}, \tag{3}$$

where $\nu \in \mathbb{R}^{B \times c \times (hwt)}$ represents the waypoint time step embedding, and $\delta$ denotes the Multilayer Perceptron (MLP) network that extracts trajectory information. $g$ is then embedded into the input sequence of the diffusion transformer and processed together with the token information $R_{re}^{(\text{emb})}$.

**Diffusion Transformer.** We developed a diffusion-based world model to learn from and generate within the latent space $R_m$, while integrating trajectory labels $T_r$ and denoising time steps $v_d$ as control conditions. In the model diffusion learning process, we constructed a forward noise process that gradually introduces noise to the latent space $R_m$: $q\left(R_{\text{re}}^g | R_{\text{re}}\right) = N\left(R_{\text{re}}^g; \sqrt{\overline{\sigma^g}} R_{\text{re}}, \left(1 - \overline{\sigma^g}\right) I\right)$, where the constant $g$ represents the embedding of trajectories and time steps. Utilizing the reparameterization trick, we can sample: $R_{\text{re}}^g = \sqrt{\overline{\sigma^g}} R_{\text{re}} + \sqrt{1 - \overline{\sigma^g}} \varepsilon^g$, where $\varepsilon^g \sim N(0, I)$. The 4D occupancy diffusion model is trained to learn the reverse propagation process. To invert the forward process corruption:

$$p_\theta\left(R_{\text{re}}^{g-1} | R_{\text{re}}^g\right) = N\left(\mu_\theta\left(R_{\text{re}}^g\right), \Sigma_\theta\left(R_{\text{re}}^g\right)\right), \tag{4}$$

where neural networks predict the statistical properties of $p$. By reparameterizing as a noise prediction network, the model can be trained using the simple mean squared error between the predicted noise $\hat{R}_{re}^g$ and the sampled gaussian noise $R_{re}^g$: $L_{simple}(\theta) = \frac{1}{2}(\hat{R}_{re}^g - R_{re}^g)^2$. However, to train the diffusion model with learned reverse process covariance, the full KL divergence term needs to be optimized. We follow diffusion models approach (Dhariwal and Nichol, 2021): train first with $L_{simple}(\theta)$, then with the full $L$. Once $p$ is trained, new token can be sampled by initializing $R_{re}^g \sim N(0, I)$ and sampling $R_{re}^{g-1} \sim p(R_{re}^{g-1} | R_{re}^g)$ using the reparameterization trick.

Overall, tokens $R_m$ processed in the initial stage as $R_{re}$ are passed to a series of transformer blocks for further refinement. These blocks effectively capture the relationships between trajectory information and tokens. Regarding noisy image input processing, the diffusion transformer employs specific attention mechanisms and loss functions to minimize the impact of noise on model performance, ensuring robust operation in noisy environments. To incorporate trajectory labels $T_r$ and denoising time steps $v_d$ as additional control conditions, we feed them as supplementary inputs alongside token embeddings into the transformer blocks. This enables the model to dynamically adjust its processing based on these conditions, thereby better adapting to various trajectory control requirements. In the end, the trained diffusion-based world model successfully transforms pure noise and trajectory labels $T_r$ into $T_o \in \mathbb{R}^{B \times c \times h \times w \times t}$, which are eventually decoded into $R_o$ through the 3D decoder.

# 4 EXPERIMENTS

As the first 4D occupancy world model in the field of autonomous driving, OccSora offers a deeper understanding of the relationship between autonomous driving scenes and vehicle trajectories without requiring any input of 3D bounding boxes, maps, or historical information. It can construct a long-time sequence world model that adheres to physical laws. We have conducted a series of quantitative experiments and visualizations to illustrate this.

## 4.1 IMPLEMENTATION DETAILS

Our experiments are conducted on the widely used nuScenes-Occupancy dataset (Caesar et al., 2020), which is currently one of the most mainstream and standard datasets, supporting many well-known research studies (Hu et al., 2023b; Wang et al., 2023c). For the OccSora-Base network, we applied three rounds of compression to 32 consecutive frames and increased its channel dimension to 128. Subsequently, we conducted further comparative and ablation experiments under different components and trajectory scenarios. We trained using the AdamW optimizer with an initial learning rate set to $1 \times 10^{-5}$ and a weight decay of 0.01. Using 8 NVIDIA GeForce A100 GPUs, we set a batch size of 2 per GPU. For the training of the 4D occupancy scene tokenizer, we needed about 42GB of memory per GPU to train for 150 epochs, which took 50.6 hours. For the diffusion-based world model, we needed about 47GB of memory per GPU to train for 1,200,000 steps, which took 108 hours. We used the Fréchet distance between the spatial-temporal latent representations, referred to as FOD, as the evaluation metric for the generation.

## 4.2 4D OCCUPANCY RECONSTRUCTION

The compression and reconstruction of 4D occupancy are essential for learning the latent spatiotemporal correlations and features necessary for image generation. Unlike traditional models for video and image processing, OccSora operates one dimension higher than occupancy maps for single frames and two dimensions higher than images. Therefore, achieving efficient compression and accurate reconstruction is paramount. Figure 5 depicts the ground truth and reconstruction of the occupancy. We also conducted a quantitative analysis of 4D occupancy reconstruction. OccSora outperforms

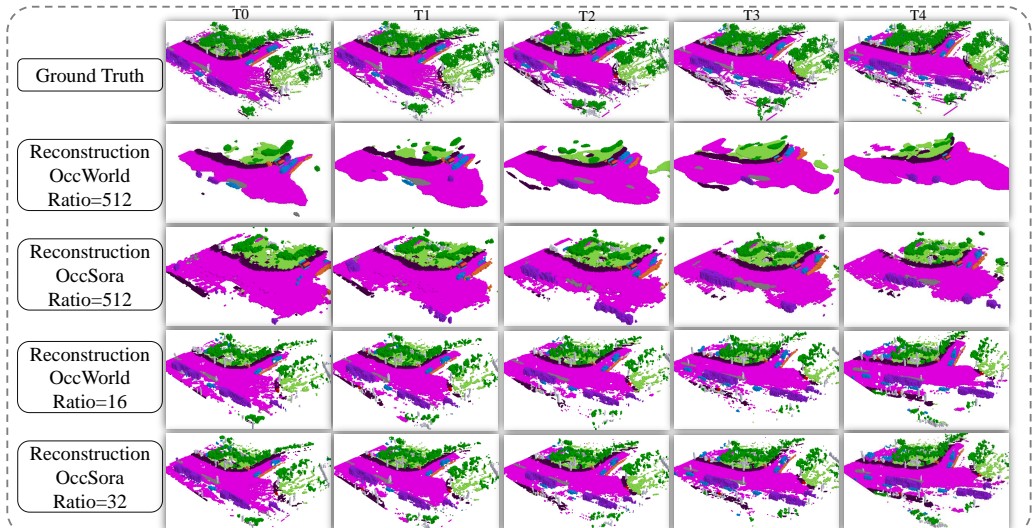

Figure 5: **Visualization of reconstruction of the 4D occupancy scene tokenizer.** OccSora outperforms OccWorld under the same compression conditions, whether at a high compression ratio of 512x or a low compression ratio of 16x.

Table 1: **The quantitative analysis of 4D occupancy reconstruction.** We compressed OccSora and OccWorld at a 16x compression ratio (input dimensions: 200x200x6, output dimensions: 50x50x6) and at a 512x compression ratio (input dimensions: 200x200x32, output dimensions: 25x25x4). In terms of both the high compression ratio of 512x and the low compression ratio of 16x, the quantitative results of OccSora outperform those of OccWorld under the same compression conditions.

| Method | Ratio | IoU | mIoU | Others | barrier | bicycle | bus | car | const. veh. | motorcycle | pedestrian | traffic cone | trailer | truck | drive. suf. | other flat | sidewalk | terrain | man made | vegetation |
|---|---|---|---|---|---|---|---|---|---|---|---|---|---|---|---|---|---|---|---|---|
| OccWorld | 16 | 62.2 | 65.7 | 45.0 | 72.2 | **69.6** | 68.2 | 69.4 | 44.4 | 70.7 | 74.8 | **67.6** | 54.1 | 65.4 | 82.7 | 78.4 | 69.7 | 66.4 | 52.8 | 43.7 |
| OccSora | 16 | **71.9** | **69.0** | **47.7** | **73.5** | 62.8 | **79.6** | **72.2** | 50.1 | **73.7** | **78.9** | 41.7 | **63.8** | **73.8** | **84.4** | **83.6** | **77.3** | **75.6** | **54.1** | **48.5** |
| OccWorld | 512 | 22.4 | 8.2 | 2.0 | 12.1 | 0.0 | 4.4 | 3.9 | 0.3 | 0.0 | 0.5 | 1.2 | 0.3 | 2.2 | 48.8 | 19.3 | 20.2 | 17.6 | 4.0 | 4.3 |
| OccSora | 512 | 37.0 | 27.4 | 11.7 | 22.6 | 0.0 | 34.6 | 29.0 | 16.6 | 8.7 | 11.5 | 3.5 | 20.1 | 29.0 | 61.3 | 38.7 | 36.5 | 31.1 | 12.0 | 18.4 |

Figure 6: **The visualization of the gradual generation of accurate scenes sequence as the model undergoes iterative training.**

the existing method (OccWorld (Zheng et al., 2023)) by a large margin under both high and low compression rates. This is because OccSora further considers temporal interactions for 4D occupancy reconstruction, which is important in the autonomous driving scenario, as shown in Table 1.

### 4.3 4D OCCUPANCY GENERATION

In the diffusion-based world model for the 4D occupancy generation task, we used tokens generated by the OccSora model, trained with 32 frames, as input for our generation experiments. In Figure 6, we present the visual results of across training iterations, from 10,000 to 1,200,000 steps. These visual results indicate that as the number of training iterations increases, the accuracy of the OccSora model continuously improves, demonstrating the generation of coherent scenes.

**Trajectory Video Generation.** OccSora has the capability to generate various dynamic scenes based on different input trajectories, thus learning the relationship between ego vehicle trajectories and scene evolution in autonomous driving. As shown in Figure 7, we input different vehicle trajectory motion patterns into the model, demonstrating the 4D occupancy for go straight, turning right, and motionless. We conducted experiments at different scales for generating trajectories, revealing that

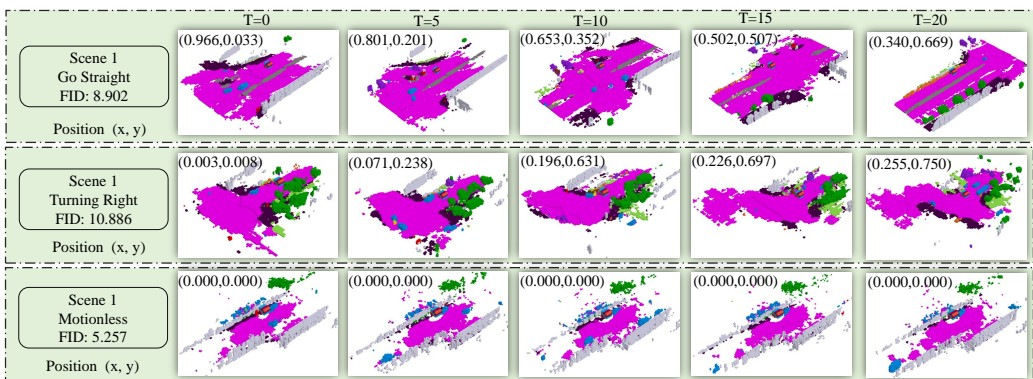

Figure 7: **4D occupancy generation under different input trajectories.** From top to bottom, there is go straight, turning right, and motionless, with each scene generation corresponding to the trajectory, ensuring logical coherence and continuity.

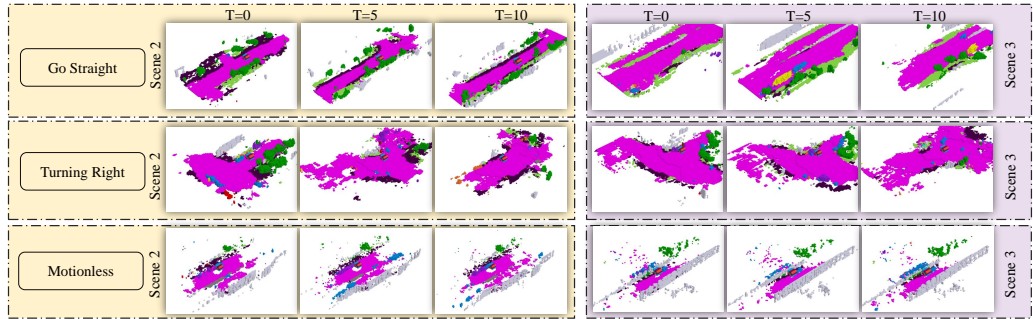

Figure 8: **Generating diverse continuous scenes under trajectory control.** The generated scenes exhibit diversity while maintaining the stability of the original trajectory control.

Table 2: **Results of ablative evaluation on different components.** We quantitatively evaluated the impact of different compression rates, components, and channel dimensions on the reconstruction and generation results through controlled variables.

| Input Size | Token Size | Channel | Class | way. embed. | Trajectory | IoU | mIoU | FOD |
|---|---|---|---|---|---|---|---|---|
| 32x200x200 | 128x4x25x25 | 8 | ✓ | ✓ | ✓ | **37.03** | **27.42** | **8.34** |
| 32x200x200 | 128x4x25x25 | 8 | ✓ | ✗ | ✓ | **37.03** | **27.42** | 87.26 |
| 32x200x200 | 128x4x25x25 | 8 | ✓ | ✓ | ✗ | **37.03** | **27.42** | 17.48 |
| 32x200x200 | 128x4x25x25 | 4 | ✓ | ✓ | ✓ | 29.67 | 23.21 | 34.24 |
| 32x200x200 | 128x8x50x50 | 8 | ✓ | ✓ | ✓ | 32.91 | 24.4 | 72.32 |
| 12x200x200 | 64x3x50x50 | 8 | ✓ | ✓ | ✓ | 26.73 | 14.12 | 187.78 |
| 12x200x200 | 64x3x25x25 | 8 | ✓ | ✓ | ✓ | 22.423 | 9.274 | 270.23 |
| 12x200x200 | 32x3x25x25 | 8 | ✓ | ✓ | ✓ | 13.595 | 3.847 | 465.18 |

the FOD score is lowest for stationary scenes and higher for curved scene, indicating the complexity of continuously modeling curved motion scenes and the simplicity of modeling stationary scenes.

**Scene Video Generation.** Diversity in scenes is crucial under reasonable trajectory control. We tested the reconstruction of 4D occupancy scenes for different scenarios under three trajectories to verify the generalization performance of generating scenes under controllable trajectories. In Figure 8, the left and right parts respectively demonstrate the capability to generate different scenes under the same trajectory. In the reconstructed scenes, surrounding trees and road environments exhibit random variations while still maintaining the logic of the original trajectory, showcasing the model's ability to maintain robustness in generating scenes corresponding to the original trajectory amidst its generalization across different scenarios.

### 4.4 ABLATION AND ANALYSIS

**Analysis of the Tokenizer and Embeddings.** We conducted an ablation of the proposed components including different compression scales, the number of class tokenizer discretizations, waypoint embed-

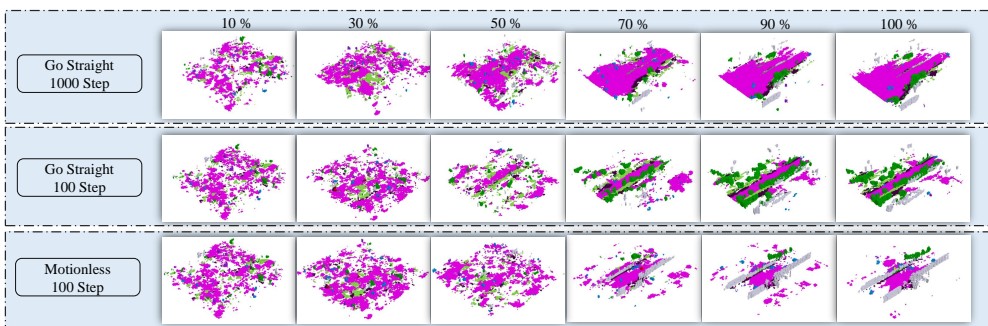

Figure 9: **The denoising ratios under different trajectories or denoising steps.** Denoising steps and trajectories have a minor impact on the quality of generation, while denoising ratios have a significant effect.

Table 3: **The quantitative analysis of different scales regarding denoising steps and denoising rates.** Denoising steps have a relatively minor impact on the model, whereas denoising rates and model scales significantly affect the quality of the generated outputs.

| Step | Input Size | Token Size | FOD | | | | | |
|------|-----------|-----------|------|------|------|------|------|------|
| | | | 10% | 30% | 50% | 70% | 90% | 100% |
| 10 | 32x200x200 | 128x4x25x25 | 49863 | 34927 | 17630 | 339 | 42 | 9.1 |
| 100 | 32x200x200 | 128x4x25x25 | 53297 | 29521 | 19471 | 1084 | 72 | 10.08 |
| 1000 | 32x200x200 | 128x4x25x25 | 32171 | 10284 | 5924 | 591 | 17 | 8.94 |
| 10 | 12x200x200 | 64x3x50x50 | 71293 | 54625 | 5644 | 7416 | 742 | 431 |
| 100 | 12x200x200 | 64x3x50x50 | 81274 | 53431 | 45346 | 3161 | 456 | 446 |
| 1000 | 12x200x200 | 64x3x50x50 | 43631 | 33415 | 17431 | 4366 | 379 | 353 |

dings, and vehicle trajectory embeddings, as shown in Table 2. When the number of class tokenizer discretizations was reduced from 8 to 4, the reconstruction accuracy dropped by approximately 18%. The FOD score also declined after removing the waypoint embeddings. Without position embeddings, the generated scenes lacked motion control and displayed almost linear movement patterns influenced by the data distribution. Additionally, at lower compression ratios, although the reconstruction performance was better compared to higher compression ratios, the lack of higher-dimensional feature correlations prevented the generation of effective scenes.

**Analysis of the Generation Steps.** The total number of denoising steps and the denoising rate can affect the generation quality to some extent. As shown in Figure 9, as the denoising rate increases, the generated scenes become progressively clearer. According to the quantitative results in Table 3, increasing the total number of denoising steps can improve generation accuracy to a certain extent. However, the generation quality is much more significantly influenced by the model's token size and the number of channels than by the total number of denoising steps.

**Analysis of the 4D occupancy Reconstruction.** To understand the effectiveness of temporal modeling, we further conduct ablation experiments on OccSora with different compression strategies. We analyze the effect of different compression ratios. We observe that the reconstruction performance generally improves as the compression ratio decreases. This is reasonable since a lower compression rate results in less information loss yet poses more challenges (e.g., taking up more memory) for generation. We therefore select the largest compression rate (512) for long occupancy video generation, as shown in Table 4. This unified temporal compression effectively captures the dynamic changes of various elements, improving long-sequence modeling capabilities compared to progressive autoregressive methods.

**Analysis of the 4D occupancy prediction.** We compared and quantitatively evaluated our proposed OccSora model against other models. We follow existing methods to employ a history of 2 seconds to predict occupancy for the upcoming 3 seconds. We compare OccSora with existing methods including 4D OCC (Khurana et al., 2023), Copilot4D (Zhang et al., 2023a), and OccWorld (Zheng et al., 2023) and report mIoU and IoU at different time stamps in the Table 5. We also provide visualizations in Figure 10. We observe major improvements of our OccSora over existing methods, demonstrating the superiority of spatial-temporal 4D occupancy modeling and diffusion-based future generation.

Table 4: **The quantitative analysis of 4D occupancy reconstruction of different compression ratios.** We conducted an ablation study on the reconstruction performance of OccSora under different compression ratios, gradually increasing from a lower compression ratio (16x) to a higher ratio (512x). The quantitative results show that as the compression ratio increases, the reconstruction performance gradually decreases.

| Method | Ratio | IoU | mIoU | Others | barrier | bicycle | bus | car | const. veh. | motorcycle | pedestrian | traffic cone | trailer | truck | drive. suf. | other flat | sidewalk | terrain | man made | vegetation |
|---|---|---|---|---|---|---|---|---|---|---|---|---|---|---|---|---|---|---|---|---|
| OccSora | 16 | **71.9** | 69.0 | 47.7 | **73.5** | 62.8 | **79.6** | **72.2** | 50.1 | 73.7 | 78.9 | 41.7 | **63.8** | **73.8** | 84.4 | **83.6** | **77.3** | **75.6** | 54.1 | 48.5 |
| OccSora | 32 | 65.6 | **69.4** | **57.6** | 72.4 | **75.4** | 71.1 | 70.5 | **53.1** | **76.0** | **79.2** | **78.8** | 59.9 | 68.6 | **84.6** | 81.2 | 73.0 | 69.5 | **56.1** | **48.8** |
| OccSora | 64 | 54.5 | 39.6 | 26.4 | 34.8 | 0.0 | 57.9 | 50.0 | 41.5 | 18.7 | 35.0 | 8.1 | 27.9 | 51.6 | 69.7 | 63.2 | 59.1 | 53.1 | 27.0 | 33.2 |
| OccSora | 128 | 45.8 | 29.9 | 15.7 | 23.9 | 0.0 | 47.0 | 39.1 | 34.0 | 0.0 | 15.3 | 0.4 | 24.0 | 43.3 | 47.2 | 46.7 | 45.7 | 41.4 | 19.6 | 28.5 |
| OccSora | 256 | 40.9 | 27.1 | 22.9 | 29.8 | 0.0 | 50.5 | 44.1 | 38.4 | 0.0 | 17.2 | 0.1 | 19.6 | 47.4 | 42.0 | 42.7 | 51.5 | 47.0 | 24.9 | 30.8 |
| OccSora | 512 | 37.0 | 27.4 | 11.7 | 22.6 | 0.0 | 34.6 | 29.0 | 16.6 | 8.7 | 11.5 | 3.5 | 20.1 | 29.0 | 61.3 | 38.7 | 36.5 | 31.1 | 12.0 | 18.4 |

Table 5: **The comparison of OccSora with other models focuses on occupancy prediction using historical occupancy as a condition for future generations.** Following existing methods. We set the compression ratio to match the OccWorld configuration (processing each frame separately and compressing from 200x200 to 50x50) and employ a 2-second history to predict occupancy for the following 3 seconds. Quantitative results show that OccSora outperforms other models.

| Method | Input | mIoU (%) | | | | | IoU (%) | | | | |
|---|---|---|---|---|---|---|---|---|---|---|---|
| | | 0s | 1s | 2s | 3s | Avg. | 0s | 1s | 2s | 3s | Avg. |
| 4D OCC | 4D-Occ | 66.38 | 15.66 | 11.85 | 8.78 | 12.10 | 62.29 | 26.79 | 20.88 | 17.93 | 21.87 |
| Copilot4D | 4D-Occ | 66.38 | 20.32 | 13.56 | 9.34 | 14.41 | 62.29 | 29.45 | 22.57 | 18.28 | 23.43 |
| OccWorld | 4D-Occ | 66.38 | 25.78 | 15.14 | 10.51 | 17.14 | 62.29 | 34.63 | 25.07 | 20.18 | 26.63 |
| OccSora | 4D-Occ | **66.97** | **32.77** | **22.04** | **14.40** | **23.07** | **68.78** | **41.39** | **33.68** | **29.97** | **35.01** |

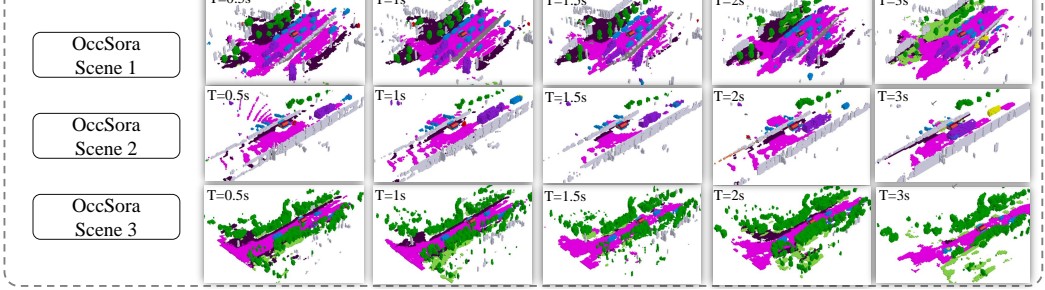

Figure 10: **Prediction results of OccSora by using past frames as conditions.** We follow existing methods to employ a history of 2 seconds to predict occupancy for the upcoming 3 seconds.

## 5 CONCLUSION AND LIMITATIONS

In this paper, we have introduced a framework for generating 4D occupancy to simulate 3D world development in autonomous driving. Using a 4D scene tokenizer, we obtain compact representations for input and achieve high-quality reconstruction for long-sequence occupancy videos. Then, we learn a diffusion transformer on the spatiotemporal representations and generate 4D occupancy conditioned on a trajectory prompt. Through experiments on the nuScenes dataset, we demonstrate accurate scene evolution. In the future, we will investigate more refined 4D occupancy world models and explore the possibilities of end-to-end autonomous driving under closed-loop data environments.

**Limitations.** The advantage of a 4D occupancy world model lies in its ability to establish an understanding of the relationship between scenes and motion. However, due to limitations in the granularity of voxel data within the dataset, constructing more finely detailed 4D scenes is not feasible. Moreover, using the real-world occupancy of the first frame to explore long-term 4D occupancy prediction tasks with controllable trajectories is an important direction for future research.

# 6 CODE OF ETHICS AND ETHICS STATEMENT

All authors of this submission have read and adhered to the ICLR Code of Ethics. We acknowledge compliance with the ethical standards in all aspects of this work, including submission, review, and discussion. This research follows ethical practices concerning data privacy, fairness, and transparency. There are no conflicts of interest, and no involvement of human subjects or harmful applications.

# 7 REPRODUCIBILITY STATEMENT

We have made efforts to ensure the reproducibility of our results. Relevant sections of the paper, appendix, and supplementary materials provide clear descriptions of the model, algorithms, datasets, and data processing steps. Anonymous source code will also be provided as part of the supplementary materials to facilitate reproducibility.

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

# 8 APPENDIX

## 8.1 TRAINING DETAILS

In Table 6, we provide detailed information about each step of our OccSora-Base model. The bolded sections indicate the dimensions of the model's intermediate outputs at the 4D occupancy scene tokenizer stage and the inputs and generated outputs of the diffusion-based world model. Similarly, for OccSora-Small and OccSora-Tiny, we also present the model's input and compression sizes, as shown in Table 7. Additionally, the reconstruction accuracy at each frame is also displayed in Table 8.

Table 6: **Basic structure of the OccSora-Base network.** The OccSora-Base network illustrates the design and dimensional changes of critical steps in two stages. The bolded sections indicate the flow of tokens from the intermediate 4D occupancy scene tokenizer stage into the diffusion-based world model, and the output of the diffusion-based world model feeding back into the 4D occupancy scene tokenizer stage.

| Model | Step | Input dimension | Layer | Output dimension |
|---|---|---|---|---|
| OccSora-Base | First Step | 32x200x200x16 | Class Embedding | 32x200x200x16x8 |
| OccSora-Base | First Step | (16x8)x32x200x200 | 3D Encoder Down Sample 1 | 256x100x100x16 |
| OccSora-Base | First Step | 256x100x100x16 | 3D Encoder Down Sample 2 | 256x50x50x8 |
| OccSora-Base | First Step | 256x50x50x8 | 3D Encoder Down Sample 3 | 512x25x25x4 |
| OccSora-Base | First Step | 512x25x25x4 | 3D Encoder Attention | 256x25x25x4 |
| OccSora-Base | First Step | 256x25x25x4 | 3D Encoder Attention | **128x25x25x4** |
| OccSora-Base | First Step | **128x25x25x4** | 3D Decoder Up Sample 1 | 256x50x50x8 |
| OccSora-Base | First Step | 256x50x50x8 | 3D Decoder Up Sample 2 | 256x100x100x16 |
| OccSora-Base | First Step | 256x100x100x16 | 3D Decoder Up Sample 3 | 128x200x200x32 |
| OccSora-Base | First Step | 128x200x200x32 | Reshape Out | 32x200x200x16x8 |
| OccSora-Base | Second Step | **128x25x25x4** | Reshape | 128x2500 |
| OccSora-Base | Second Step | 128x2500 | Pos Embedding | 128x2500 |
| OccSora-Base | Second Step-Other | 32x2 | MLP | 128x2500 |
| OccSora-Base | Second Step-Other | Time Embedding | 128x2500 | |
| OccSora-Base | Second Step-Fuse | 128x2500 | Fuse | 128x2500 |
| OccSora-Base | Second Step | 128x2500 | DiT(FN-Attention) | 128x2500 |
| OccSora-Base | Second Step | 128x2500 | Reshape | **128x25x25x4** |

Table 7: **Three different SoraOcc network structures and their compression dimensions.** In OccSora-base, we present the model with the best performance, featuring 32 frames and 128 layers of channels. On the other hand, OccSora-Small and OccSora-Tiny showcase models with 12 frames and channel sizes of 64 and 32 layers, respectively.

| Method | Step | Original size | Token size | Reshape size |
|---|---|---|---|---|
| OccSora-Base | First Step | 32x200x200 | 128x4x25x25 | - |
| OccSora-Small | First Step | 12x200x200 | 64x3x25x25 | - |
| OccSora-Tiny | First Step | 12x200x200 | 32x3x25x25 | - |
| OccSora-Base | Second Step | - | 128x4x25x25 | 128x2500 |
| OccSora-Small | Second Step | - | 64x3x25x25 | 128x1875 |
| OccSora-Tiny | Second Step | - | 32x3x25x25 | 128x1875 |

Table 8: **The scene reconstruction capabilities of OccSora models across various frames.** We demonstrate the accuracy of three OccSora models in reconstruction tasks across different frames.

| Method | Metric | Original size | Token size | Type | 0S | 1S | 2S | 3S | 4S | 5S | 6S | 7S | 8S | 9S |
|---|---|---|---|---|---|---|---|---|---|---|---|---|---|---|
| OccSora-Base | IoU | 12x200x200 | 64x3x25x25 | 3D | 32.9 | 37.0 | 37.4 | 35.4 | 36.0 | 37.4 | 37.8 | 35.9 | 36.1 | 37.3 |
| OccSora-Small | IoU | 32x200x200 | 128x4x25x25 | 3D | 22.4 | 26.1 | 27.8 | 28.7 | 29.3 | 28.8 | 27.0 | 25.8 | 26.2 | 27.5 |
| OccSora-Tiny | IoU | 12x200x200 | 32x3x25x25 | 3D | 13.5 | 14.8 | 15.8 | 16.6 | 17.4 | 18.0 | 18.1 | 18.3 | 18.4 | 18.5 |
| OccSora-Base | mIoU | 12x200x200 | 64x3x25x25 | 3D | 23.0 | 27.4 | 27.3 | 25.3 | 25.8 | 27.7 | 27.2 | 25.4 | 25.6 | 27.4 |
| OccSora-Small | mIoU | 32x200x200 | 128x4x25x25 | 3D | 9.2 | 11.2 | 13.3 | 14.4 | 15.2 | 14.2 | 12.4 | 11.4 | 11.6 | 12.3 |
| OccSora-Tiny | mIoU | 12x200x200 | 32x3x25x25 | 3D | 3.8 | 4.3 | 4.7 | 5.1 | 5.4 | 5.6 | 5.6 | 5.6 | 5.6 | 5.6 |

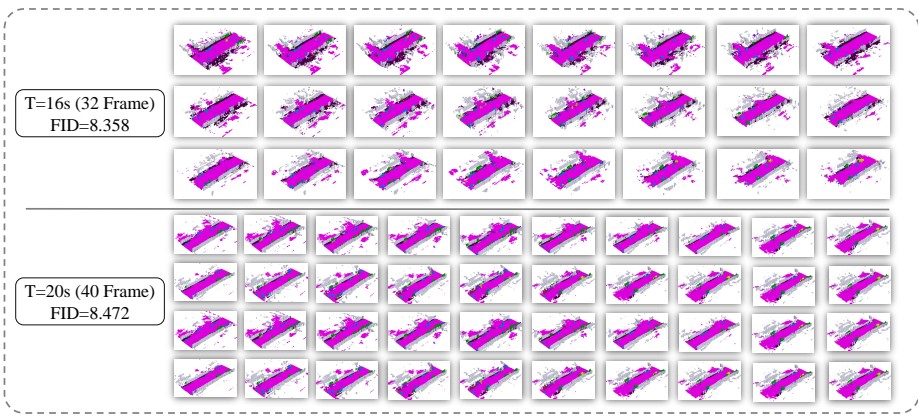

Figure 11: **Visualizations of generating longer sequences(20seconds).** Scenes with a full duration of 20 seconds were selected for training to demonstrate the model's ability to generate extended 4D occupancy. The model produces consistent high-quality scenes with a comparable FOD score.

Table 9: **Performance of OccSora on the Occ3D-Waymo.** We tested the proposed algorithm on the Occ3D-Waymo, where (CRLO) represents the categories of curb, road, lane marker, and other ground.

| Method | Ratio | IoU | mIoU | car | pedestrian | sign | CYCLIST | traiffic light | pole | construction cone | bycycle | motorcycle | building | vegetation | trunk | CRLO | walkable, sidewalk | unobsrvd | vis |
|---|---|---|---|---|---|---|---|---|---|---|---|---|---|---|---|---|---|---|---|
| OccSora-waymo | 16 | 71.2 | 65.6 | 79.8 | 70.4 | 57.0 | 69.2 | 57.1 | 58.2 | 55.4 | 61.4 | 65.7 | 83.6 | 52.1 | 47.3 | 86.6 | 79.1 | 78.9 | 49.2 |
| OccSora-waymo | 512 | 71.2 | 65.6 | 79.8 | 70.4 | 57.0 | 69.2 | 57.1 | 58.1 | 55.2 | 61.4 | 65.4 | 83.6 | 52.1 | 47.8 | 86.5 | 79.1 | 78.9 | 49.2 |

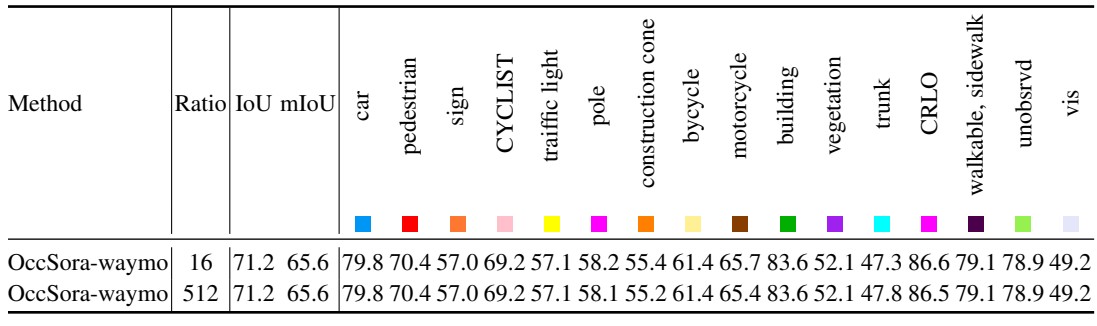

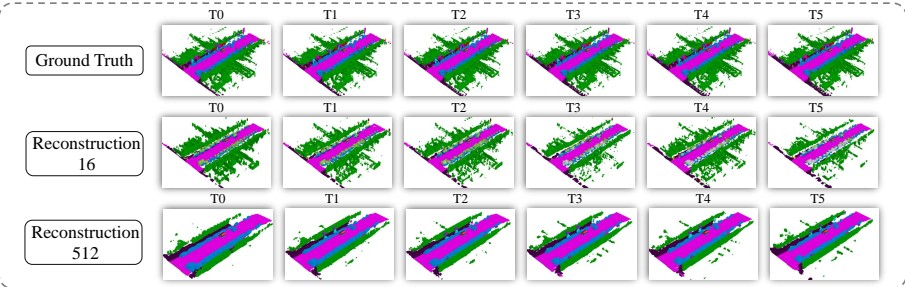

Figure 12: **Figure shows the visualization results of our model's predictions on the Occ3D-Waymo.** From top to bottom, the rows represent the ground truth, predictions under 16x compression, and predictions under 512x compression, all conducted under the same experimental conditions as OccSora.

## 8.2 GENERATING LONGER SEQUENCES

Our framework can readily generalize to longer sequences by modifying the input size of the 4D occupancy scene tokenizer. We selected 16 seconds due to the limitations of the nuScenes data. We have selected scenes with the full length of 20 seconds for training to demonstrate our model's ability to generate longer 4D occupancy, as shown in Figure 11. We see that our model still generates consistent scenes with high quality and comparable FOD. We will test our model on longer sequences like 40 seconds with newly released data.

Table 10: **Prediction comparison between the autoregressive and diffusion-based methods.** It can be observed that the autoregressive method experiences an increase in inference time as the number of frames grows, while the generative model maintains consistent inference time due to its parallel processing capability. This highlights the temporal efficiency advantage of OccSora, particularly for long-sequence generation tasks.

| Model | 1 Frame | 8 Frames | 16 Frames | 32 Frames |
|---|---|---|---|---|
| OccWorld | 27 ms | 220 ms | 431 ms | 855 ms |
| OccSora | 49 ms | 52 ms | 51 ms | 57 ms |

Table 11: **The experimental settings and their corresponding mIoU values are summarized.** It shows how the input size, model size, and mIoU change with the number of frames under a fixed compression ratio and sampling iterations.

| Frame | Sampling Iterations | Compression Ratio | Input Size | Model Size | mIoU (%) |
|---|---|---|---|---|---|
| 1 | 50 | 64 | 200x200x1 | 25x25x1 | 33.96 |
| 8 | 50 | 64 | 200x200x8 | 25x25x8 | 19.71 |
| 16 | 50 | 64 | 200x200x16 | 25x25x16 | 13.68 |
| 32 | 50 | 64 | 200x200x32 | 25x25x32 | 8.42 |

### 8.3 GENERALIZATION ACROSS OCC3D-WAYMOS

To analyze the differences in model performance across different datasets, we conducted training and testing on the Occ3D-Waymo (Tian et al., 2024), with visualization results shown in Figure 12. The quantitative results in Table 9 demonstrate the model's high accuracy on the Occ3D-Waymo, showcasing its robustness and adaptability.

### 8.4 TEMPORAL-SPATIAL EFFICIENCY ANALYSIS

OccSora operates as a diffusion-based generative framework and OccWorld functions as an autoregressive approach; the two exhibit substantial differences in generation efficiency. OccWorld generates data sequentially for each timestep, resulting in a total generation time of $T_{AR} = T \cdot t_{AR}$ with a time complexity of $O(T)$, implying that generation time increases linearly with the sequence length $T$. In contrast, OccSora generates the entire sequence in parallel through denoising steps, with a total generation time of $T_{GM} = N \cdot t_{GM}$ and a time complexity of $O(1)$. Here, the number of denoising steps $N$ is typically much smaller than $T$. As a result, when $T \gg N$, OccSora demonstrates significantly higher temporal efficiency than OccWorld, especially in tasks requiring long-sequence generation. Quantitative analyses are provided in Table 10. Additionally, considering that the test time varies depending on the experimental setup, we provide detailed explanations of the experimental configurations in Table 11. This includes the number of sampling iterations, compression ratio, input dimensions, model dimensions, and evaluation metrics.

### 8.5 PREDICTION COMPARISON WITH OCCWORLD

We present a visual comparison of OccSora and OccWorld on the prediction task, as shown in Figure 13. From top to bottom, the figure displays the ground truth followed by the prediction results of the two models.

### 8.6 DIVERSE DATA BALANCING

We separated the straight-driving and turning trajectories in the NuScenes dataset (Caesar et al., 2020)and balanced the data volumes between the two categories. The visualization 14 demonstrate that using the balanced dataset leads to improved performance in predicting steering trajectories.

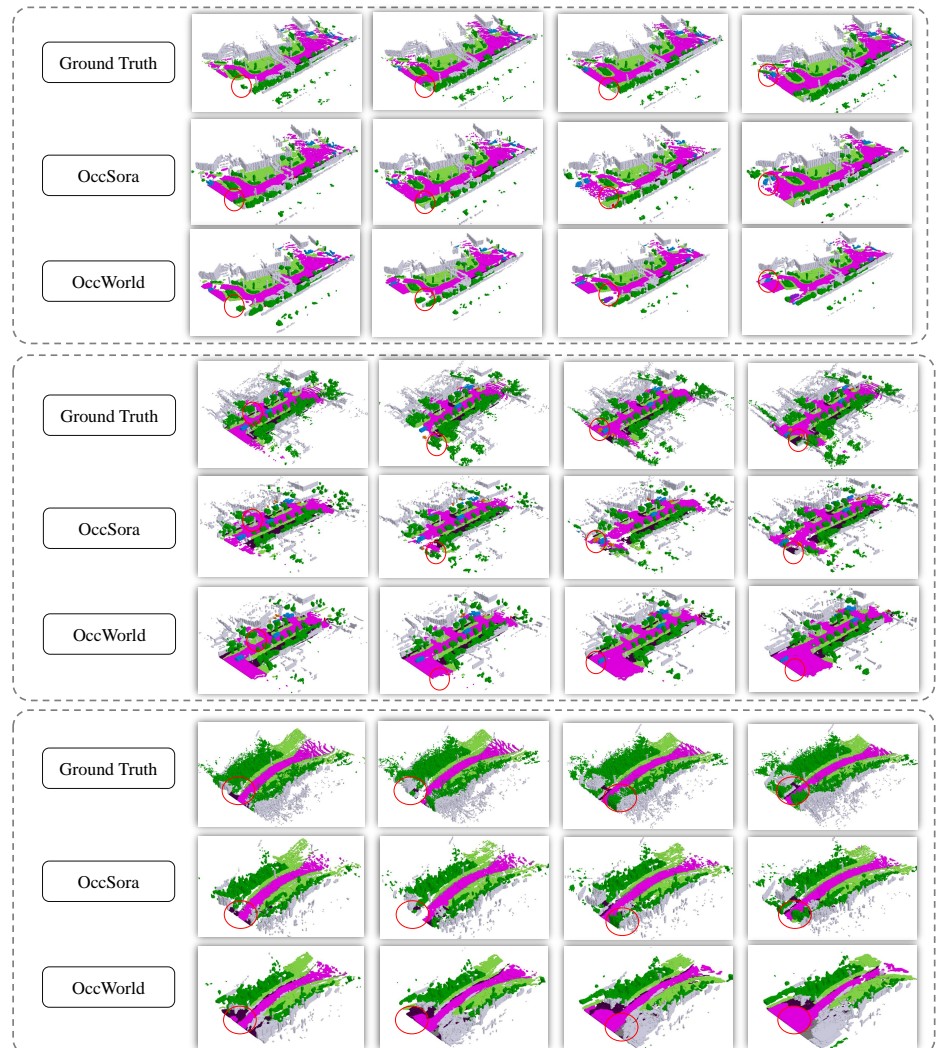

Figure 13: **Provides a visual comparison between OccSora and OccWorld on the prediction task.** From top to bottom, show the ground truth and the prediction results of the two models. The visualization highlights the differences in the predictive capabilities of each method under the same experimental conditions.

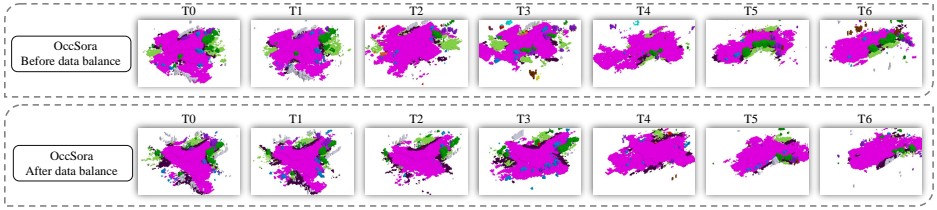

Figure 14: **The difference in generation quality before and after balancing the dataset.** The improved quality of the generated trajectories demonstrates the effectiveness of the balanced dataset in enhancing model performance.

## 8.7 QUANTIZERS ABLATION STUDY

To explore the relationship between quantizers, a codebook in the tokenizer, and the continuous compression model, we conducted an ablation study. In the VAE experiments, we removed the VQ

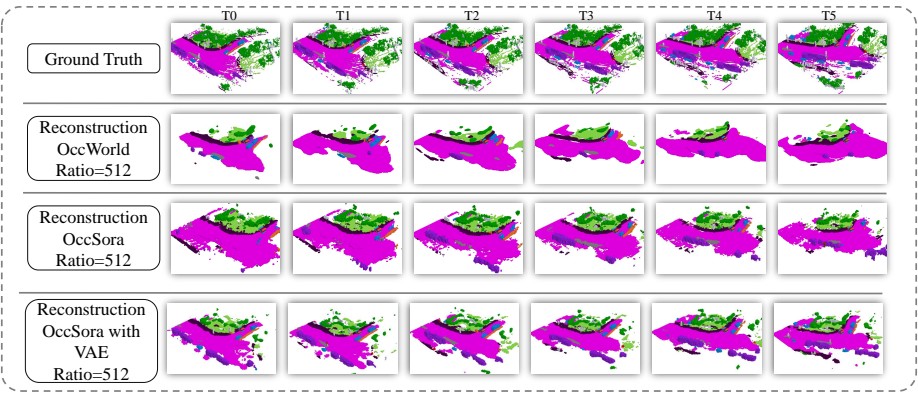

Figure 15: **The visualization results of the VAE experiment after removing the VQ and codebook.** The image demonstrates the effectiveness of the approach in reconstructing 4D Occupancy, with visual comparisons showing its performance.

Table 12: **The table presents the quantitative results of the VAE experiment, highlighting the performance of the approach in reconstructing 4D Occupancy.** The numerical values indicate the accuracy and effectiveness of the method under the experimental conditions.

| Method | Ratio | IoU | mIoU | Others | barrier | bicycle | bus | car | const. veh. | motorcycle | pedestrian | traffic cone | trailer | truck | drive. suf. | other flat | sidewalk | terrain | man made | vegetation |
|---|---|---|---|---|---|---|---|---|---|---|---|---|---|---|---|---|---|---|---|---|
| OccWorld | 512 | 22.4 | 8.2 | 2.0 | 12.1 | 0.0 | 4.4 | 3.9 | 0.3 | 0.0 | 0.5 | 1.2 | 0.3 | 2.2 | 48.8 | 19.3 | 20.2 | 17.6 | 4.0 | 4.3 |
| OccSora | 512 | 37.0 | 27.4 | 11.7 | 22.6 | 0.0 | 34.6 | 29.0 | 16.6 | 8.7 | 11.5 | 3.5 | 20.1 | 29.0 | 61.3 | 38.7 | 36.5 | 31.1 | 12.0 | 18.4 |
| OccSora-VAE | 512 | 43.8 | 31.7 | 15.7 | 37.2 | 13.4 | 46.4 | 37.4 | 23.9 | 26.8 | 20.9 | 6.4 | 27.3 | 37.1 | 69.0 | 46.9 | 45.2 | 41.0 | 21.3 | 23.9 |

and codebook, and performed experiments under the same conditions. The visualization results are shown in Figure 15, and the quantitative results are presented in Table 12. The results demonstrate that both approaches effectively reconstruct 4D Occupancy.

## 8.8 OCCSORA MODEL VISUALIZATION

To provide results for longer time series, we comprehensively present the generated scenes under different ego vehicle trajectory controls, namely Go Straight, Turning Right, Motionless, and Accelerate, as depicted in Figure 17. Additionally, we also showcase the equivalent control methods under different scenes in Figure 16. Furthermore, dynamic demonstrations are available in the accompanying video.

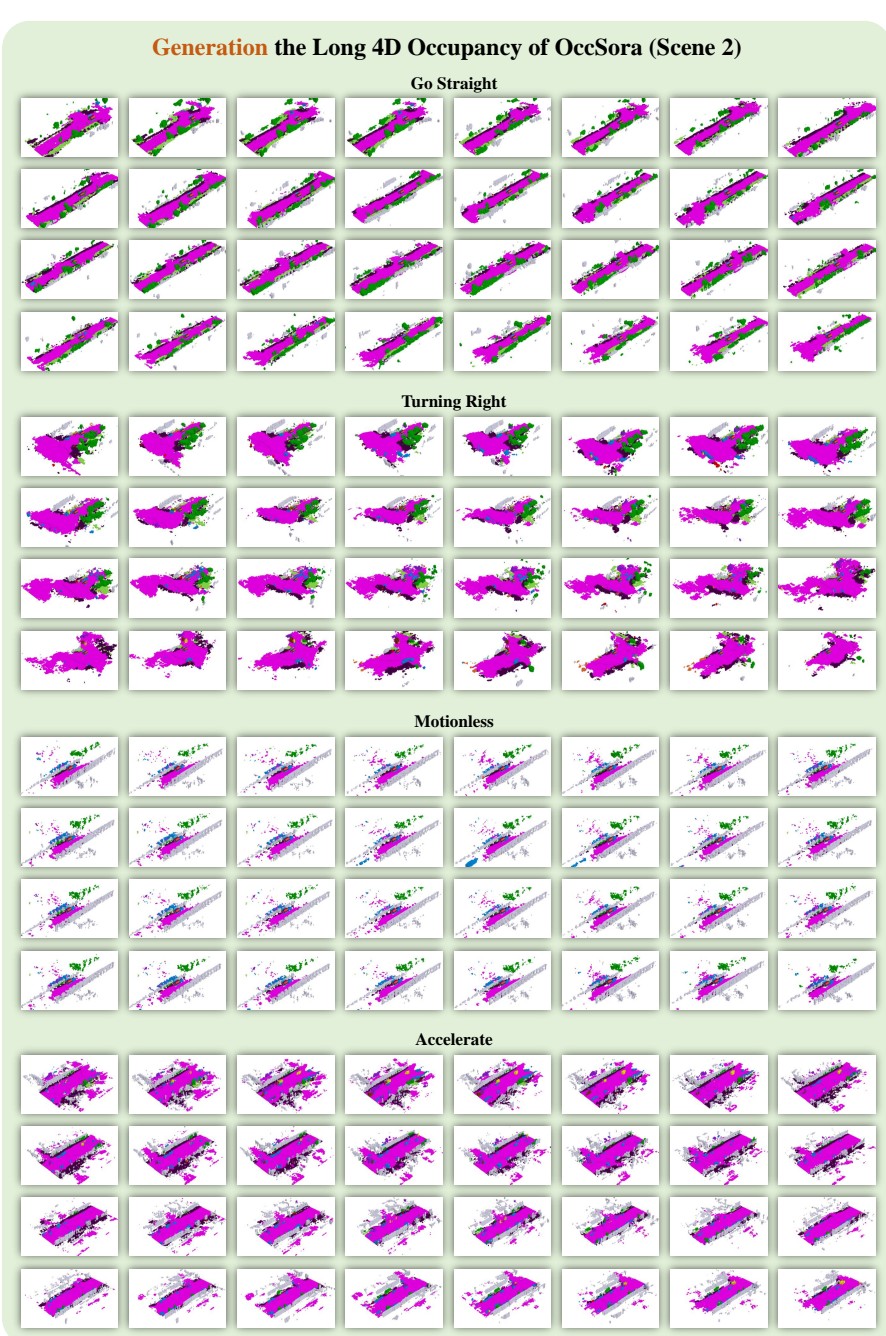

Figure 16: **The generalization ability to generate different scenes under fixed ego vehicle trajectories.** From top to bottom, we respectively showcase the capabilities of generating different scenes under the four vehicle trajectories: Go Straight, Turning Right, Motionless, and Accelerate.

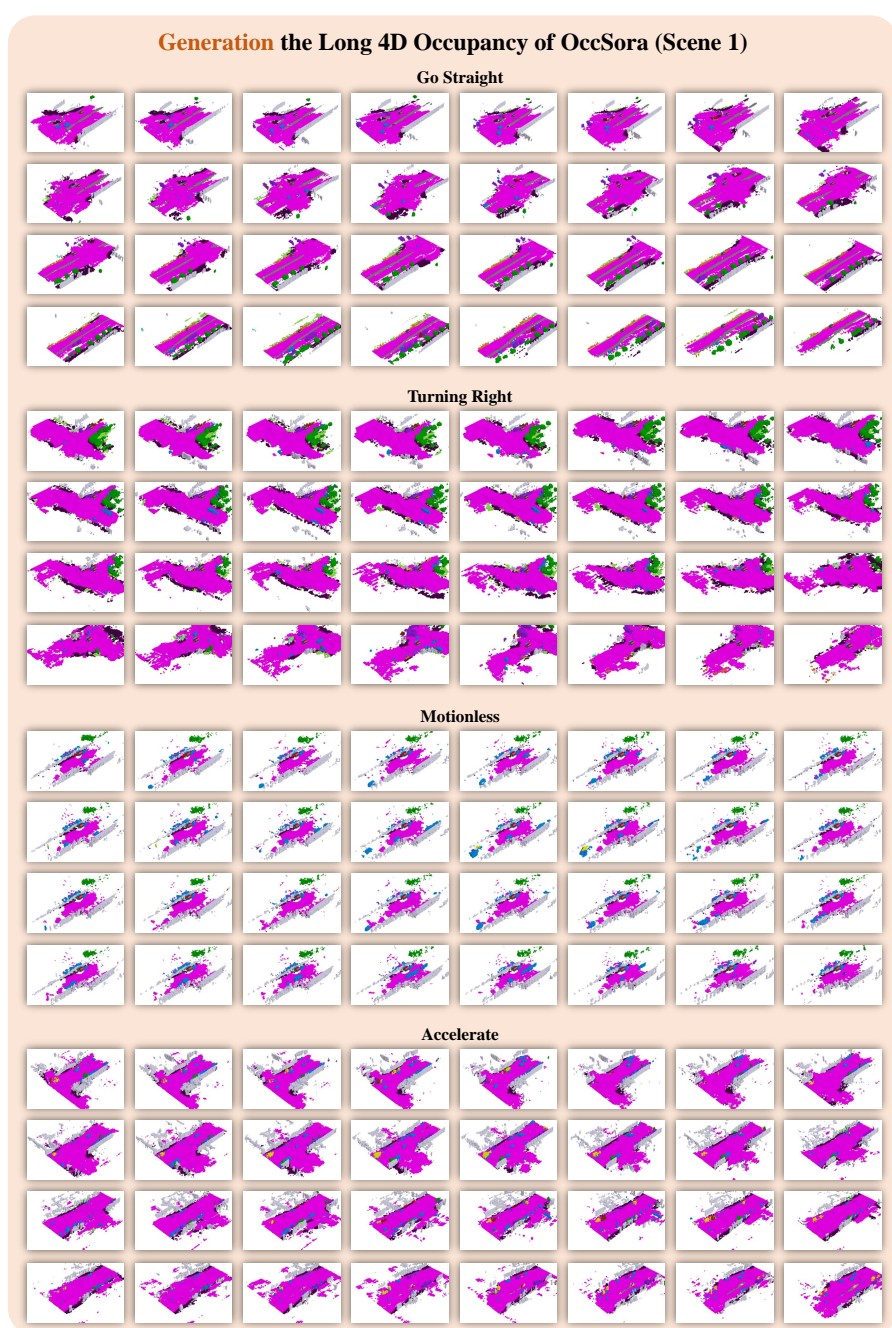

Figure 17: **The ability to generate long time-series 4D occupancy under different trajectory controls.** From top to bottom, we present long-term continuous scenes generated under four types of ego vehicle trajectories: Go Straight, Turning Right, Motionless, and Accelerate.

