# OpenReview forum: "OccSora: 4D Occupancy Generation Models as World Simulators for Autonomous Driving"
_ICLR.cc/2025/Conference — Submitted to ICLR 2025_

### Official Review · Reviewer_WBnQ · 2024-10-23

**Soundness:** 3
**Presentation:** 1
**Contribution:** 3
**Rating:** 5
**Confidence:** 5

**Summary:**

This paper introduces **OccSora**, a 4D occupancy generation model designed to capture the evolution of 3D scenes. Notably, it is **the first generative 4D occupancy world model for autonomous driving**. The model employs a 4D scene tokenizer to create compact spatiotemporal representations from 4D occupancy inputs, facilitating the high-quality reconstruction of long-sequence occupancy videos. Additionally, OccSora incorporates a diffusion transformer to generate 4D occupancy conditioned on trajectory prompts. Extensive experiments on the nuScenes dataset demonstrate OccSora's ability to generate 16-second videos with realistic 3D layouts and temporal consistency, highlighting its capacity to comprehend dynamic driving scenes. As a result, OccSora holds promise as a simulation tool for autonomous driving decision-making.

**Strengths:**

1. The paper is well-organized, making it easy to follow.
2. This paper introduces the first generative 4D occupancy world model for autonomous driving and proposes a new generation task for occupancy data, which is a contribution to the community.
3. The design of the 4D tokenizer (Section 3.2) is particularly novel. The experiments demonstrate its effectiveness in preserving 3D occupancy geometry with temporal consistency, even under high compression rates.
4. The ablation study is thorough and comprehensive.

**Weaknesses:**

1. I don't see the necessity of using quantizers and a codebook in the tokenizer (line 200). Since the authors employ a diffusion-based model, typically paired with a continuous compression model (e.g., VAE), this differs from the auto-regressive models (e.g., OccWorld) where such techniques might be more fitting. Could the author clarify their reasoning or provide supporting empirical results?
2. The idea of compressing the latent spatiotemporal dimensions together is interesting. However, in the diffusion model, simply flattening the tokens results in temporal modeling inefficient. Why not adopt an additional temporal layer, as is common in video-based diffusion models? Some references that might be helpful include:
  - Align Your Latents: High-Resolution Video Synthesis with Latent Diffusion Models
  - Scaling Latent Video Diffusion Models to Large Datasets
  - VDT: General-purpose Video Diffusion Transformers via Mask Modeling
  - Latte: Latent Diffusion Transformer for Video Generation
3. The generation metric is not described in sufficient detail. Firstly, FID is designed to evaluate images, yet this paper generates occupancy videos. At the very least, the authors should consider using FVD. Additionally, no details or references explain how FID is adapted for occupancy data or how occupancy features are extracted from pre-trained networks, given that this is the first work to generate 4D occupancy.
4. I am confused by the two experimental setups in Section 4.3. For the "Trajectory Video Generation" experiment (line 406), I expected to see how the generation would vary with different trajectory inputs. However, Figure 7 presents three entirely different scenes, even in the first frame. In the "Scene Video Generation" experiment (line 413), the authors claim they use the same trajectory for each motion case (line 417), but what causes the differences in the left and right parts of Figure 8? Is it the random seed or the input latents? The inference process is not explained at all.
5. I have concerns about the 4D occupancy prediction experiment described in line 484. The task is to forecast future frames based on historical frames. However, the authors use a 3D encoder with no masking strategy, which means the encoder can "see" the future frames, making the prediction task unfair. This could be proved by the results in Table 8 of the appendix, where the reconstruction IoU is higher for the 1-9s but lower at 0s, since there is no historical information for the 0s frame. Furthermore, the paper only demonstrates the process starting from pure noise. How is historical occupancy input into the DIT model?

**Questions:**

1. **The main text of the paper exceeds the maximum page limit of 10 pages**, as per the ICLR 2025 guidelines. Please check the details at ICLR Call for Papers.
2. The authors should consider using `\citep{}` in the paper for better formatting. Refer to the ICLR LaTeX template for the appropriate usage and differences between citation commands.
3. In the caption of Table 1, the output dimension for the "512x" setting should be listed as $ 25 \times 25 \times 4 $.
4. In Figure 6, the model name should be updated to **OccSora-base**.

---

> ### Author Response · Authors · 2024-11-24
> **Response to Reviewer WBnQ (Part I)**
>
> ## Weakness:
>
> ### W1:  Continuous Compression Model (VAE) Experiment Results.
>
> Thank you for your suggestion. Based on your feedback, we have included experiments using the VAE model. Compared to the VQ-VAE method, which relies on quantizers and a codebook, the VAE method simplifies the generation process. The experimental results (see Section 8.7) show that the VAE method performs well and even surpasses the VQ-VAE method. The use of VQ-VAE was primarily aimed at compressing the Occupancy for efficient generation tasks. Both methods can achieve this goal, and we believe this does not affect the core contribution of our approach. Notably, our method is compatible with both VAE and VQ-VAE models. We have updated the manuscript to include this additional experiment.
>
>
> Quantitative results of the VAE experiment on 4D Occupancy reconstruction. The numerical values indicate the accuracy and effectiveness of the method under the experimental conditions.
>
> | Method       | Ratio | IoU  | mIoU | Others | Barrier | Bicycle | Bus | Car | Const. Veh. | Motorcycle | Pedestrian | Traffic Cone | Trailer | Truck | Drive. Suf. | Other Flat | Sidewalk | Terrain | Man Made | Vegetation |
> |--------------|-------|------|------|--------|---------|---------|-----|-----|-------------|------------|------------|--------------|--------|-------|-------------|------------|---------|--------|----------|------------|
> | **OccWorld** | 512   | 22.4 | 8.2  | 2.0    | 12.1    | 0.0     | 4.4 | 3.9 | 0.3         | 0.0        | 0.5        | 1.2          | 0.3    | 2.2   | 48.8        | 19.3       | 20.2    | 17.6   | 4.0      | 4.3        |
> | **OccSora**  | 512   | 37.0 | 27.4 | 11.7   | 22.6    | 0.0     | 34.6 | 29.0 | 16.6        | 8.7        | 11.5       | 3.5          | 20.1   | 29.0  | 61.3        | 38.7       | 36.5    | 31.1   | 12.0     | 18.4       |
> | **OccSora-VAE** | 512 | 43.8 | 31.7 | 15.7   | 37.2    | 13.4    | 46.4 | 37.4 | 23.9        | 26.8       | 20.9       | 6.4          | 27.3   | 37.1  | 69.0        | 46.9       | 45.2    | 41.0   | 21.3     | 23.9       |
>
>
> ### W2: Temporal Attention and Space-Time Learning.
>
> Thank you for your insightful suggestion. As you mentioned, utilizing temporal attention layers for video generation is a common approach in general video generation tasks. In the works you referenced and other related studies, video generation typically involves two-stage training: first, training an image generation model, and then using the pre-trained image generation model with added temporal attention layers to handle video tasks. Alternatively, some methods use images as inputs to generate videos. Our task differs as it takes only the trajectory as input (analogous to text input in generation tasks) without providing the first frame (analogous to the initial image). Thus, we chose to simultaneously learn spatiotemporal features during the compression and reconstruction stages. This approach leverages the strong spatiotemporal correlation inherent to 4D occupancy, reducing the cost and complexity of learning spatial and temporal features simultaneously in the generation network. It provides a more direct method compared to generating the first frame and then expanding it into a video, as our model learns spatiotemporal relationships during compression.
>
> ### W3: Clarification of FID and FVD Metrics
>
> We apologize for not explaining this point clearly. The FID and FVD metrics mentioned here are not entirely equivalent to image metrics. Instead, they refer to metrics for the latent space in occupancy tasks after spatiotemporal compression, specifically the FID calculated on the tokens of the 4D Occupancy after compression using VAE, which we call FOD. We have updated the last sentence in Section 4.1 to clarify this distinction more accurately.

---

> ### Author Response · Authors · 2024-11-24
> **Response to Reviewer WBnQ (Part II)**
>
> ## Weakness:
>
> ### W4: Clarification of Generation Task and Control Trajectory.
>
> We apologize for any confusion caused by our explanation. Our generation task differs from prediction tasks based on the current occupancy. OccSora focuses on trajectory-based 4D occupancy generation to enhance the understanding of the relationship between autonomous driving trajectory planning and scene transformations, enabling the creation of new controllable scenes. This is analogous to text-to-video generation in image tasks. The figure primarily illustrates the capability to generate diverse 4D occupancy scenes under the same trajectory condition by varying random seeds, ensuring generalization while adhering to the control trajectory constraints.
>
> ### W5: Fair Comparison with OccWorld and Compression Rate Clarification.
>
> Thank you for your suggestion, and we apologize for not clearly explaining this point. To ensure a fair comparison with OccWorld, we used the same compression rate as OccWorld, and therefore, we did not apply compression under 3D conditions (as you noted, this would lead to information leakage). The results in Appendix Table 8 are from experiments involving 3D reconstruction without per-frame information extraction. In our occupancy prediction task, we use the ground truth of the first two frames and noise as input, applying cross-attention to learn conditional information and predict the next three frames. We deeply appreciate your invaluable feedback.
>
> ## Questions:
>
> ### Q1: Clarification on Page Limit for Ethical Sections.
>
> Thank you for bringing this to our attention. After reviewing the [ICLR Author Guide](https://iclr.cc/Conferences/2025/AuthorGuide), we confirm that the Code of Ethics, Ethics Statement, and Reproducibility sections are not counted toward the page limit but must not exceed one page. We appreciate your reminder.
>
> ### Q2: Revision of Citation Format.
>
> Thank you for your suggestion. We have revised all citation formats to use **\citep{}**. Your feedback is greatly appreciated.
>
> ### Q3: Corrections and Thorough Review.
>
> Thank you for your valuable suggestion. We have made the necessary corrections and conducted a thorough review.
>
> ### Q4: Manuscript corrections.
>
>  Thank you for your valuable suggestion. We have updated the model name in Figure 6 to "OccSora-base" and conducted a thorough review of the relevant sections to ensure clarity.

---

### Official Review · Reviewer_e9CJ · 2024-10-28

**Soundness:** 3
**Presentation:** 2
**Contribution:** 2
**Rating:** 6
**Confidence:** 2

**Summary:**

This paper proposes OccSora, a diffusion-based 4D occupancy generation model designed for autonomous driving. It utilizes a 4D scene tokenizer to get spatial-temporal representations, allowing for high-quality reconstruction of long-term occupancy videos. The model generates 4D occupancy data based on trajectory prompts, generating videos with 3D layouts and temporal consistency. Experiments demonstrate OccSora's potential as a world simulator for autonomous driving.

**Strengths:**

1. This paper is well-organized, presenting a clear flow making it easy to read.
2. This paper proposes a generative 4D occupancy world model designed for autonomous driving, along with a novel generation task for occupancy data.
3. The experiments provide an evaluation of the model's capabilities and performance across various scenarios.

**Weaknesses:**

1. The quality of the turning cases shown in Figure 8 and Figure 13 appears to be quite poor. The motion trend is barely visible, and the scene structure beside the road becomes corrupted in the later frames. This undermines the claim of trajectory awareness. I suspect this issue is related to the imbalanced data distribution in the nuScenes dataset, which could present an interesting challenge. Unfortunately, the paper offers no further analysis or solution regarding this problem.
2. The authors spent a lot of space discussing the diffusion transformer (lines 269–290). However, much of this appears to replicate the contribution of the original DIT paper. This section could be shortened or moved to a preliminary section instead.
3. The mathematical notation throughout the paper is quite disorganized. Some of the symbols are unusual, and the authors should consider using more standard notation (e.g., what is "mi" in $ R_{mi} $?). The excessive use of superscripts and subscripts makes it difficult to follow. Additionally, the authors should avoid using unnecessary symbols for unimportant variables (e.g., $ x $ in line 246). There are also several instances of misused symbols. For example:
  1. The symbol $ N $ is used multiple times for different meanings, such as a label (line 176), nearest code operation (line 205), and Gaussian noise (line 271).
  2. In line 254, the positional encoding notation is inconsistent with Equation (2).
  3. In Equation (3), the $ t $ in $ v(t) $ should represent the diffusion timestep, but it is confused with the occupancy timestep $ t $.
  4. In line 265, $ v $ is referred to as the waypoint timestep embedding, but it is suddenly called the denoising timestep in line 269.

4. There are several grammar issues and typos:
  1. In line 181, "represents" should be "representing."
  2. In line 197, "model ability" should be "model's ability."
  3. In line 253, "model understanding" should be "model's understanding."
  4. In line 266, it should read: " $ g $ is then embedded into the input sequence ..."
  5. In line 280, "occuancy" should be "occupancy."
  6. In line 525, "reprort" should be "report."

**Questions:**

1. In Table 2, it's unclear why increasing the resolution (from $ 128 \times 4 \times 25 \times 25 $ to $ 128 \times 8 \times 50 \times 50 $) improves the reconstruction IoU and mIoU, but results in a decrease in generation FID. This discrepancy needs further explanation.
2. The figures showing occupancy data throughout the paper are quite small, and the resolution is low, making it difficult to see details. The authors should consider placing fewer samples in each row and enlarging the figures for better visibility.

---

> ### Author Response · Authors · 2024-11-24
> **Response to Reviewer e9CJ**
>
> ## Weakness:
>
>
> ### W1: Balanced Dataset for 4D Occupancy.
>
> We agree with your insights and suggestions. In response, we reconstructed the dataset to make it more balanced and supplemented our experiments. The results, presented in Section 8.6, demonstrate that a more balanced dataset significantly benefits steering reconstruction. Thank you for such an excellent suggestion!
>
> ### W2: Section Shortening for Conciseness.
>
>  Thank you very much for your suggestion. We have shortened this section to make the paper more concise and focused.
>
> ### W3: Simplification of Notation in Equations and Figures.
>
> Thank you for your suggestion. We apologize for the overuse of annotations in the equations. We have simplified $R_{mi}$ (compressed occupancy) to $R_{m}$ , $R_{in}$  (ground truth input) to $R_{i}$ , and $R_{tr}$  (vehicle trajectories) to $R_{r}$ . These changes have also been applied to the figures to ensure consistency and avoid unnecessary symbols for less significant variables. Regarding your comment on line 246, we thoroughly reviewed it but could not locate the $x$  you mentioned. If you could provide more specific details, we would be happy to address them in the revised version.
>
> ### W4: Revision of Type Label and Code Notation.
>
> Thank you for your insightful suggestion, which indeed highlighted a potential source of confusion. We have revised all relevant instances and renamed type label to $T_l$ and code $N$ to $N_c$. All equations and references involving these terms have also been updated, and we conducted a thorough review of the entire manuscript to ensure consistency.
>
> ### W5: Correction of Positional Encoding Notation.
>
> Thank you for your meticulous suggestion. We have corrected positional encoding to $emb$ throughout the paper.
>
>
> ### W6: Update of Notation for Diffusion Time Steps.
>
> Your suggestion is excellent. To clearly distinguish between different instances of $t$, we have updated the notation for diffusion time steps to $t_d$.
>
> ### W7: Revision of Denoising Time Steps Terminology
>
> Thank you, and we apologize for the incorrect terminology. We have revised denoising time steps to  $\nu _d$ and checked all related instances in the equations to ensure accuracy.
>
> ### W8-W14: Clarity.
>
> We are deeply grateful for your detailed and diligent review. Your meticulous feedback has been immensely valuable. We have addressed all the points you raised and made the corresponding revisions. Once again, we sincerely thank you for your careful and thoughtful suggestions.
>
>
>
> ## Questions:
>
> ### Q1: Balancing Token Count for Reconstruction and FID.
>
> Thank you very much for your suggestion. The issue primarily arises from changes in the number of tokens. The token count needs to be carefully balanced—not too large, nor too small. A larger token count leads to an increase in the generated space, which in turn increases the complexity and difficulty of generating high-quality content. While a higher token count can improve reconstruction accuracy, it also adds more content for the model to generate, which can worsen the FID, indicating lower generation quality. This happens because the model might focus on adding unnecessary details or become less precise, which impacts the overall visual quality. Therefore, it's crucial to maintain a higher reconstruction rate while selecting a smaller number of tokens to balance the generation quality and avoid introducing additional complexity that could hurt performance.
>
> ### Q2: Adjustments to Sample Size and Figure Size.
>
> Thank you for your suggestion. We have reduced the number of samples and enlarged the figures in the main text to better meet your requirements and expectations.

---

### Official Review · Reviewer_Y4yg · 2024-10-31

**Soundness:** 2
**Presentation:** 2
**Contribution:** 2
**Rating:** 5
**Confidence:** 5

**Summary:**

The paper introduces a trajectory-aware 4D occupancy generation model capable of understanding ego car trajectories and enabling trajectory-controllable scene generation.

**Strengths:**

This paper proposes a trajectory-aware 4D occupancy generation model, which can comprehend the trajectories of ego car and realize trajectory-controllable scene generation. The model uses a 4D scene tokenizer to create compact, discrete spatial-temporal representations of 4D occupancy inputs, achieving high-quality reconstructions for long-sequence occupancy videos. The proposed OccSora can generate long-term occupancy videos up to 16 seconds.

**Weaknesses:**

1. This paper claims in the abstract and introduction that existing autoregressive methods "suffer from inefficiency to model long-term temporal evolutions". It is unclear how the authors address this, as the multiple denoising steps of the diffusion model can also be time-consuming. I suggest that the authors provide some **evidences to demonstrate that the proposed architecture is more efficient**, such as runtime comparisons or theoretical complexity analyses between the proposed approach and existing autoregressive methods.
2. The experiment on the 4D occupancy prediction is not convincing. The metrics indicate that OccSora significantly outperforms OccWorld; however, the **generation quality** in qualitative results is not good enough. It is beneficial to provide side-by-side visual comparisons between OccSora and baseline methods, highlighting specific areas where the generation quality differs.
3. It is unclear **how 4D occupancy prediction is conducted** in Fig. 5 and Tab. 10, as there appears to be no frame condition in the proposed architecture. I suggest that the authors provide a more detailed explanation of how the model handles frame conditioning for 4D occupancy prediction in the proposed architecture.
4. As shown in Fig.12, the model can generate different scenes with different motion conditions. However, their first frames are different. As far as the reviewer knows, if the authors want to **claim the model is controllable**, the model should be able to generate different future predictions with the same initial frame(s) using different trajectories. I recommend that the authors provide additional experiments or qualitative results to demonstrate this .
5. Why are the IoU and mIoU values at 0s in Table 5 different from the reconstruction results in Table 1? Both seem to measure reconstruction performance.

**Questions:**

The reviewer has identified five major concerns and would like the authors' responses to these points. Please answer each concern in the rebuttal stage. The reviewer will respond according to the authors' rebuttal in the discussion phase.

---

> ### Author Response · Authors · 2024-11-24
> **Response to Reviewer Y4yg**
>
> ## Weakness:
>
> ### W1:  Autoregressive and Diffusion Model Inference Time.
>
> Thank you for your suggestion. We have carefully analyzed this issue in Section 8.4 and supplemented it with additional experiments. Specifically, the inference time of autoregressive models grows linearly with the number of time steps. In contrast, the denoising steps in diffusion models are fixed and remain unaffected by the number of time steps. We greatly appreciate your valuable feedback.
>
> Prediction comparison between the autoregressive and diffusion-based methods. It can be observed that the autoregressive method experiences an increase in inference time as the number of frames grows, while the generative model maintains consistent inference time due to its parallel processing capability. This highlights the temporal efficiency advantage of OccSora, particularly for long-sequence generation tasks.
>
> | Model       | 1 Frame | 8 Frames | 16 Frames | 32 Frames |
> |-------------|---------|----------|-----------|-----------|
> | **OccWorld** | 27 ms   | 220 ms   | 431 ms    | 855 ms    |
> | **OccSora**  | 49 ms   | 52 ms    | 51 ms     | 57 ms     |
>
>
> ### W2:  Visualizations of OccWorld with Generation Quality Differences.
>
> Thank you for your suggestion. We have incorporated visualizations of OccWorld in Section 8.5, where we highlight some regions with differences in generation quality using red circles. While both models perform well, OccSora shows a slight improvement in prediction accuracy and completeness for the target.
>
> ### W3: Explanation of 4D Occupancy Prediction.
>
> Thank you for your suggestion, and we apologize for not explaining this point clearly. In our 4D occupancy prediction task, we use the ground truth of the first two frames as conditions, combined with noise as input. Within the model, we apply cross-attention between the noise and conditions to learn essential conditional information and output predictions for the next three frames.
>
> ### W4: 4D Occupancy with Control Mechanisms.
>
> Thank you very much for your suggestion. OccSora focuses on trajectory-based 4D occupancy generation (analogous to text-to-video generation) to enhance the understanding of the relationship between autonomous driving trajectory planning and scene transformations, enabling the creation of new controllable scenes. We recognize the importance of your suggestion and plan to further design and train the network to implement the function for 4D occupancy extension (similar to image-to-video generation), where different trajectories and occupancy of the first frame can serve as input to generate continuous 4D occupancy.
>
> ### W5: Comparison with OccWorld in Compression Settings.
>
> Thank you very much for your suggestion. To ensure a fair comparison with OccWorld, we set the compression rate to match OccWorld's settings (processing each frame separately and compressing 200x200 to 50x50). As a result, the reconstruction metrics differ from those in Table 1. This discrepancy arises due to differences in compression rates and experimental settings.
>
> Thank you very much for each of your suggestions.

---

> > ### Comment · Reviewer_Y4yg · 2024-11-25
> > **Official Comment by Reviewer Y4yg**
> >
> > I appreciate the authors' response; however, my concerns have not been fully addressed.
> >
> > 1. Regarding Q1, the testing time varies depending on the experimental settings. The authors should provide a more detailed explanation covering: (1) the number of sampling iterations, (2) the compression ratio, (3) the input size, (4) the model size, and (5) the final metrics. As far as the reviewer is aware, the inference time of DiT for 512x512 resolution using DiT-XL/2 on single image generation exceeds 300 seconds. The inference times reported in the table are not convincing.
> >
> > 2. For Q2, I appreciate the authors' efforts in providing the comparison. However, the generation quality of OccWorld remains superior. For instance, in the third row, the wall in the background is smooth and realistic in OccWorld's results, whereas OccSora's outputs are highly fragmented. With only one comparison provided, this evidence is insufficient to substantiate the claims.
> >
> > 3. For Q3, as far as the reviewer knows, the baseline method uses the ground truth of the first four frames (2 seconds) as a condition and generates six frames (3 seconds). This differs from the authors' clarification.
> >
> > 4. For Q4 and Q5, I appreciate the response. Since the experimental settings across multiple tables differ, I suggest that the authors clarify these discrepancies in the manuscript.

---

> > > ### Author Response · Authors · 2024-11-26
> > > **Response to Reviewer Y4yg**
> > >
> > > ### R1: Metrics and Inference Time
> > >
> > > Thank you very much for your suggestions. We highly appreciate and agree with your feedback. These metrics are indeed necessary, and we have supplemented them in Section 8.4 of the manuscript, with the corresponding table provided below. Your observations are correct—DiT takes over 300 seconds for inference on 512x512 resolution. In our case, due to using a significantly smaller number of tokens (Tx25x25), the inference speed has increased, resulting in a reduction in inference time.
> > >
> > > The experimental settings and their corresponding mIoU values are summarized below. They demonstrate how the input size, model size, and mIoU change with the number of frames under a fixed compression ratio and sampling iterations.
> > >
> > > | Frame       | Sampling Iterations | Compression Ratio | Input Size    | Model Size   | mIoU(%) |
> > > |-------------|---------------------|-------------------|---------------|--------------|---------|
> > > | **1 Frame** | 50                  | 64                | 200x200x1     | 25x25x1      | 33.96   |
> > > | **8 Frames**| 50                  | 64                | 200x200x8     | 25x25x8      | 19.71   |
> > > | **16 Frames**| 50                 | 64                | 200x200x16    | 25x25x16     | 13.68   |
> > > | **32 Frames**| 50                 | 64                | 200x200x32    | 25x25x32     | 8.42    |
> > >
> > >
> > >
> > > ### R2: Visual Comparison and Model Improvement
> > >
> > > Thank you very much for your suggestions; we fully agree with your perspective. In Section 8.5, we have added more visual comparison experiments across different scenarios for your review. OccSora demonstrates better prediction accuracy compared to OccWorld, which occasionally generates incomplete or incorrect scenes under certain conditions. However, as you pointed out, some of OccSora's occupancy outputs are scattered. We will address this issue and strive to improve it in our future work.
> > >
> > >
> > > ### R3: Correction
> > > We sincerely apologize for this oversight; your suggestion is absolutely correct. We used the ground truth of the first 4 frames (2 seconds) as a condition to generate 6 frames (3 seconds). The predictions for the future 6 frames (3 seconds), as shown in Figure 10, also confirm this. We will correct this error, and we deeply appreciate your feedback!
> > >
> > >
> > > ### R4/R5: Manuscript Refinements
> > > Thank you very much for your guidance. We have made further revisions to the manuscript, refining Section 5 to elaborate on its limitations in greater depth and updating Table 5 to clarify the comparative experimental setups. This ensures that your suggestions are accurately reflected in the manuscript.
> > >
> > > We greatly appreciate your diligent responses and valuable suggestions, which have helped us progressively improve and refine the manuscript. Thank you again for your guidance.

---

### Official Review · Reviewer_t69g · 2024-11-04

**Soundness:** 3
**Presentation:** 2
**Contribution:** 3
**Rating:** 6
**Confidence:** 4

**Summary:**

This paper introduces OccSora, a diffusion-based model for 4D occupancy generation in autonomous driving. Using a 4D scene tokenizer, OccSora captures compact and discrete spatiotemporal representations, allowing for the high-quality reconstruction of long-sequence occupancy videos. A diffusion transformer is subsequently trained on these representations to generate 4D occupancy, conditioned on a trajectory prompt. Experimental results on the nuScenes dataset with Occ3D occupancy annotations show that OccSora can produce 16-second videos with realistic 3D layouts and temporal coherence, demonstrating its understanding of spatial and temporal patterns in driving scenes. This method shows potential as a world simulator for decision-making in autonomous driving, addressing the inefficiencies in modeling long-term temporal evolution found in autoregressive approaches.

**Strengths:**

This paper presents OccSora, a pioneering diffusion-based 4D occupancy generation model that utilizes a 4D scene tokenizer to capture compact spatiotemporal representations, improving the modeling of long-term temporal dynamics in autonomous driving. The research is comprehensive, featuring extensive experiments on the nuScenes dataset that showcase OccSora's ability to generate realistic 16-second videos with stable 3D layouts.  The paper is clearly structured and well-written, offering thorough explanations of the model architecture, training procedure, and evaluation metrics. OccSora’s capability to generate trajectory-aware 4D occupancy scenes makes it a valuable asset for decision-making in autonomous driving, with the potential to enhance safety and efficiency.

**Weaknesses:**

The writing of this paper should be enhanced. For example,

1）Some details in the methods section (Section 3) could be presented with greater clarity. For ‘Category embedding and Tokenizer’, occupancy is represented by Rin in the text, while in Figure 3, occupancy is represented by Ro.

2）If Figures 2, Figure 3 and Figure 4 contain elements corresponding to Rin, Rmi, Ro, and Rtr, please ensure these are labeled directly within the figures.

3）Please clearly indicate the dimensional changes between features within the figure to enhance reader comprehension.

**Questions:**

Refer to Weaknesses.

---

> ### Author Response · Authors · 2024-11-24
> **Response to Reviewer t69g**
>
> ## Weakness:
>
> ### W1:  Refinement of content and images in Section 3.
>
> Thank you very much for your suggestions. We provided a more detailed description of Section 3, particularly refining parts such as the 3D Video Encoder and Category Embedding. We have redrawn Figure 3 and revised its annotations. Specifically, Ri denotes the ground truth input of 4D occupancy, and Ro represents the reconstructed output.
>
> ### W2: Clarification of Key Components.
>
> We have redrawn Figures 2, 3, and 4, adding necessary labels to clarify key components: Ri (ground truth input), Rm (compressed occupancy), Ro (reconstructed occupancy), and Rr (trajectory information).
>
> ### W3: Enhanced Figures for Dimensional Changes.
>
> We have redrawn Figures to ensure the dimensional changes between features are explicitly illustrated. Additionally, we have enlarged and bolded the text to enhance reader comprehension. We sincerely appreciate your valuable feedback. If you have any further suggestions or requirements, please do not hesitate to let us know at any time. We are eager to continue improving our paper.

---

### Official Review · Reviewer_cJwx · 2024-11-05

**Soundness:** 3
**Presentation:** 3
**Contribution:** 3
**Rating:** 5
**Confidence:** 4

**Summary:**

The paper introduces a 4D occupancy generation model called OccSora, designed to simulate the evolution of environments in autonomous driving scenarios. It proposes a 4D scene encoder to produce a compact spatiotemporal representation and uses a diffusion model to generate 4D occupancy sequences based on given trajectory prompts. OccSora addresses the inefficiency of existing autoregressive models in long-term temporal simulation, providing more physically consistent simulation support for decision-making in autonomous driving through trajectory-aware 4D generation. OccSora achieves good results on nuScenes dataset.

**Strengths:**

1.The paper is well written and organized. It’s easy to understand.
2.The authors provide abundant visualization results to show the effectiveness.

**Weaknesses:**

1. The novelty is limited. Using diffusion to generate occupancy is widely studied. And the contribution seems incremental compared to previous works in the field of occupancy prediction.
2.  It might be beneficial to conduct experiments on additional datasets, such as Waymo, to enhance the persuasiveness of the paper.
3. The text in Figure 3 is too small and unclear.

**Questions:**

See the weaknesses above

---

> ### Author Response · Authors · 2024-11-24
> **Response to Reviewer cJwx**
>
> ## Weakness:
>
> ### W1:  Controllable 4D Occupancy Generation for Autonomous Driving.
>
> Thanks for the comment. We have further reviewed related works on occupancy generation. To the best of our knowledge, there is not enough work on occupancy generation to be considered widely studied. While existing works such as DynamicCity [1] and Pyramid Diffusion [2] adopted diffusion to generate LiDAR or occupancy, they mainly focus on constructing static occupancy. However, our work is among the first to explore controllable 4D occupancy generation for autonomous driving. While occupancy prediction is one capability of OccSora, our primary focus is on trajectory-controllable 4D occupancy generation, which is critical for understanding the relationship between autonomous driving planning and scene transformations. Also, previous efforts on occupancy prediction, such as OccWorld [3], have not explored diffusion-based scene generation. Recent follow-up studies, such as OccLLaMA [4], OccVAR [5], and DOME [6] have not been published yet and should be considered concurrent work as ours.  They also fail to generate diverse 4D occupancy scenes solely based on trajectories, which limits their ability to establish a deeper understanding of the relationship between driving trajectories and scenes.
>
> For technical contributions, we propose several key designs to adapt the pipeline for 4D occupancy generation. Since per-frame occupancy resides in the 3D space, we designed a 4D occupancy scene tokenizer based on category embeddings to enable more efficient spatiotemporal encoding and decoding. Additionally, we introduce a DiT-based model for controllable 4D occupancy generation conditioned on trajectories. Specifically, we encode the trajectory as a control embedding and incorporate waypoint embeddings into the trajectory coordinates of each frame to ensure controlled scene generation. These designs significantly enhance the accuracy of generated scenes and their alignment with autonomous driving control requirements.
>
> [1]: Bian, Hengwei, et al. "DynamicCity: Large-Scale LiDAR Generation from Dynamic Scenes." arXiv preprint arXiv:2410.18084 (2024).
>
> [2]: Liu, Yuheng, et al. "Pyramid Diffusion for Fine 3D Large Scene Generation." In ECCV, 2024.
>
> [3]: Zheng, Wenzhao, et al. "Occworld: Learning a 3d occupancy world model for autonomous driving." In ECCV 2024.
>
> [4]: Wei, Julong, et al. "Occllama: An occupancy-language-action generative world model for autonomous driving." arXiv preprint arXiv:2409.03272 (2024).
>
> [5]: Jin, Bu, et al. "OccVAR: Scalable 4D Occupancy Prediction via Next-Scale Prediction."
>
> [6]: Gu, Songen, et al. "DOME: Taming Diffusion Model into High-Fidelity Controllable Occupancy World Model." arXiv preprint arXiv:2410.10429 (2024).
>
> ### W2:  Results on the Occ3D-Waymo.
>
> Thank you very much for your suggestions. We trained our model on the Occ3D-Waymo [1] and provided both quantitative and qualitative results of reconstruction and generation based on the dataset in Section 8.3 of the manuscript.
>
> We tested the proposed algorithm on the Occ3D-Waymo, where (CRLO) represents the categories of curb, road, lane marker, and other ground.
>
> | Method          | Ratio | IoU   | mIoU  | Car   | Pedestrian | Sign  | Cyclist | Traffic Light | Pole  | Construction Cone | Bicycle | Motorcycle | Building | Vegetation | Trunk | CRLO  | Walkable/Sidewalk | Unobserved | Visible |
> |------------------|-------|-------|-------|-------|------------|-------|---------|---------------|-------|-------------------|---------|------------|----------|------------|-------|-------|-------------------|------------|---------|
> | **OccSora-Waymo** | **16** | 71.2  | 65.6  | 79.8  | 70.4       | 57.0  | 69.2    | 57.1          | 58.2  | 55.4              | 61.4    | 65.7       | 83.6     | 52.1       | 47.3  | 86.6  | 79.1              | 78.9       | 49.2    |
> | **OccSora-Waymo** | **512** | 71.2  | 65.6  | 79.8  | 70.4       | 57.0  | 69.2    | 57.1          | 58.1  | 55.2              | 61.4    | 65.4       | 83.6     | 52.1       | 47.8  | 86.5  | 79.1              | 78.9       | 49.2    |
>
> [1]: Tian, Xiaoyu, et al. "Occ3d: A large-scale 3d occupancy prediction benchmark for autonomous driving." in NeurIPS 2024.
>
> ### W3: Figure 3 Redesign for Improved Clarity.
>
>  Thank you for your suggestions. We have revised Figure 3 and adjusted the font size for improved clarity and precision.

---

### Meta-Review · Area_Chair_SNJb · 2024-12-22

**Metareview:**

This paper proposed a diffusion-based 4D occupancy generation model, OccSora, for autonomous driving. Essentially, the proposed method is a trajectory conditioned occupancy generation model. I summarize strengths and weaknesses as follows.

Strengths:
1. The proposed world-model simulator task seems novel in autonomous driving.
2. Extensive results on nuScenes demonstrate consistent occupancy prediction.

Weaknesses:
1. Despite the overall claim seems novel, the techniques (diffusion for occupancy) are extensively studied in other works(e.g. DynamicCity, Pyramid Diffusion).
2. Comparisons with existing methods (e.g., OccWorld) are limited and, in some cases, favor OccSora due to differences in input and evaluation setups.
3. The paper claims a world simulator/model, however, only does occupancy prediction. It fails to provide adequate evidences of using this world simulator.

Therefore, despite several merits, the paper still fails to meet the standard of ICLR due to technical novelty, rigorous comparisons, and adequate empirical justification of the proposed method.

**Additional Comments On Reviewer Discussion:**

Initially, reviewers raised several questions regarding paper's novelty, experiment details, and paper presentations. Authors provided responses to address most of them. However, reviewers (e.g. Y4yg) still found it hard to understand experiment details even with provided responses and code. Also, reviewers' question on the usage of this proposed simulator is not addressed.

---

### Decision · Program_Chairs · 2025-01-22

Reject